# Visual tracking assessment in a soccer-specific virtual environment: A web-based study

**Alexandre Vu**[ID]⊛*, **Anthony Sorel**⊛, **Charles Faure**‡, **Antoine Aurousseau**‡, **Benoit Bideau**‡, **Richard Kulpa**⊛

Univ Rennes, Inria, M2S - EA 7470, Rennes, France

⊛ These authors contributed equally to this work.
‡ CF, AA and BB also contributed equally to this work.
* alexandre.vu@univ-rennes2.fr

**Data Availability Statement:** All relevant data are within the manuscript and its Supporting information files.

## Abstract

The ability to track teammates and opponents is an essential quality to achieve a high level of performance in soccer. The visual tracking ability is usually assessed in the laboratory with non-sport specific scenarios, leading in two major concerns. First, the methods used probably only partially reflects the actual ability to track players on the field. Second, it is unclear whether the situational features manipulated to stimulate visual tracking ability match those that make it difficult to track real players. In this study, participants had to track multiple players on a virtual soccer field. The virtual players moved according to either real or pseudo-random trajectories. The experiment was conducted online using a web application. Regarding the first concern, the visual tracking performance of players in soccer, other team sports, and non-team sports was compared to see if differences between groups varied with the use of soccer-specific or pseudo-random movements. Contrary to our assumption, the ANOVA did not reveal a greater tracking performance difference between soccer players and the two other groups when facing stimuli featuring movements from actual soccer games compared to stimuli featuring pseudo-random ones. Directing virtual players with real-world trajectories did not appear to be sufficient to allow soccer players to use soccer-specific knowledge in their visual tracking activity. Regarding the second concern, an original exploratory analysis based on Hierarchical Clustering on Principal Components was conducted to compare the situational features associated with hard-to-track virtual players in soccer-specific or pseudo-random movements. It revealed differences in the situational feature sets associated with hard-to-track players based on movement type. Essentially with soccer-specific movements, how the virtual players were distributed in space appeared to have a significant influence on visual tracking performance. These results highlight the need to consider real-world scenarios to understand what makes tracking multiple players difficult.

## Introduction

The behavior of a soccer player on the field depends on the surrounding information gathered from his/her teammates, opponents and the ball. In this complex environment, players have to

**Funding:** This study was partially funded by the ANR within the framework of the PIA EUR DIGISPORT project (ANR-18-EURE-0022) and by the Region Bretagne. The funders had no role in study design, data collection and analysis, decision to publish, or preparation of the manuscript. There was no additional external funding received for this study.

**Competing interests:** The authors have declared that no competing interests exist.

dynamically allocate their visual attention to several areas of the field in order to track the movements of others over time [1]. This ability to simultaneously track multiple targets is usually assessed in the laboratory with the Multiple-Object-Tracking (MOT) task [2] (see [3] for a tutorial review). The MOT task consists in monitoring a subset of target objects moving for a few seconds in a given space with disruptive objects of same shape and color. The number of targets that can be simultaneously tracked by an observer is limited [4], and provides an indicator of its visual tracking performance. In a resource-based theory, this limit is reached when the tracking demands consumed the observer's attentional resources [5]. Different situational features affect the attentional demands of the task such as the number of targets to track [2], the number of disruptors to abstract [6], the tracking speed [7] or the proximity between targets and disruptors [8].

Tracking performance is also conditioned by the characteristics of observers. Allen and colleagues observed an effect of expertise on tracking performance in aircraft radar operators [9]. The authors suggested that the operator advantage is associated with an automation of attentional strategies used in tracking multiple objects on screen [9]. Like radar operators, action video game players must deal with multiple objects in complex and dynamic virtual environments. Expertise effect on MOT was also observed in the video games field, where action video game players demonstrated higher tracking performance than non-video game players [10–13], and where professional video games players also demonstrated higher tracking performance than amateur video games players [14]. In addition, Green and Bavelier observed that tracking performance was improved in non-video game players after regular training on a first-person view action video game [11]. The authors suggested that this improvement in tracking performance was enabled by video game-induced improvements in some aspects of visual short-term memory [11]. In addition, playing action video games has been found to have an effect on both adult [12] and child populations [13]. We can note that age-related changes were observed in the study by Dye and Bavelier in which tracking performance increased in young people as a function of age (7–10, 11–13, 14–17, and 18–22 years) and in the study by Sekuler and colleagues in which the tracking performance was higher in young adults (20.6 years) than in older adults (75.3 years) [12, 13]. Like action video games, some sports require to attend to multiple environmental sources of information [10]. Trick and colleagues compared the tracking performance of children in and out of action sports (field hockey, soccer, ultimate, martial arts), and observed a marginal group effect in favor of action sports group [10]. Other investigations applied to sport have also shown an advantage of high-skilled athletes on the visual tracking task [1, 15–17], particularly for team sport players [18], and above all for those whose position requires extensive information gathering [16, 19]. On the contrary, Memmert and colleagues observed no difference in performance between participants with 10 years of handball practice and non-practitioners [20]. It was suggested that the tracking task proposed by Memmert and colleagues was not difficult enough to see differences between groups [17]. Qiu and colleagues observed an expertise effect when at least three targets were to be tracked, but they observed no inter-participants differences when only two targets were tracked [17]. Another explanation would be that visual tracking ability does not seem to develop linearly with the level of practice [17]. In the initial visual tracking assessment of Faubert's study [15], and in the study of Qiu and colleagues [17], differences were observed between elite athletes and intermediate athletes, but not between intermediate athletes and non-athletes. Regarding the standard of handball players in the study by Memmert and colleagues [20], it is possible that they were more of an intermediate standard than an elite standard even if they practiced for more than 10 years [21]. Finally, Memmert and colleagues also highlighted the need to use "more sport-specific attention task" to study sport expertise [20]. All of these investigations are based on generic visual stimuli and emphasize the importance of

fundamental perceptual-cognitive mechanisms to achieve high sport performance [22]. However, although the proposed task is similar to tracking the location of players on a field, the proposed stimuli are usually non sport-specific. Players do not simply monitor spheres that move randomly around the field, independently of each other, but instead seek to perceive structured collective movements using their knowledge to anticipate the evolution of the game situation [23–25]. This lack of representativeness of the visual tracking task in relation to sport-specific scenarios is concerning in two ways.

First, one might think that visual tracking performance measured in the laboratory with non-sport specific movements only partially reflects the actual ability of players to track multiple teammates and opponents moving across the field according to the logic of the game. Mann and colleagues' systematic review reported that the effect of perceptual-cognitive expertise was modulated by task representativeness [26]. Differences in visual tracking performance between participants based on their level of competition might be greater if the task tapped on more specific aspects of the studied domain of expertise [20]. Perceptual-cognitive expertise is supported by effective gaze control [26, 27], which lead Harris and colleagues to hypothesize that a different gaze strategy would have explained the superior tracking performance of participants involved in high tracking sports activity compared to participants who were not [18]. But contrary to their expectations, no differences in gaze strategy between participants were observed [18]. This may be caused by a lack of representativeness of the proposed task with respect to the constraints of the field [27]. Meyerhoff and colleagues argued for ensuring that ecological validity is maintained by "embedding the tracking task into more naturalistic scenarios" to study the effect of expertise on tracking performance [3]. This representativeness issue was also raised with respect to the question of transferring the benefits of generic perceptual-cognitive training to actual performance in the field [28–30]. For training based on a generic visual tracking task, Romeas and colleagues observed promising initial results, where soccer players' passing decision making on small-sided-games increased after training [31]. However, limitations regarding the small sample size suggested that the results needed to be replicated to confirm this conclusion [32]. Then, with a larger sample size, Harenger and colleagues also studied the transfer of benefits from visual tracking training, but they did not observe significant improvement in a soccer-specific decision making task [33]. Other contrasting results were revealed, indicating the lack of transfer of generic visual tracking training to tasks representative of actual military [34] or sporting activities [35]. This problem may stem from the lack of correspondence between the stimuli perceived in training and those in the real world [28]. Subsequently, a second concern may be raised regarding the features manipulated to stimulate the visual tracking ability.

The second concerns lies in the relationship between tracking performance and sport performance that has been studied by manipulating situational features such as the number of targets [17, 29], object velocity [1, 15, 16, 18, 19] or tracking duration [19]. Manipulating these situational features one at a time allowed to test the participants' fundamental perceptual-cognitive abilities in a visual tracking task [3, 4] and was only possible with non real-world movements scenarios. However, it is questionable whether the situational feature manipulation was consistent with the situational features that can complicate the tracking task in real-world scenarios. On the one hand, the current visual tracking training is aimed at increasing the speed threshold at which participants are able to successfully track multiple objects [15, 31, 33, 35]. Near transfer have been observed to other MOT task and mid-level transfer have been observed to working memory task [34] but, as mentioned earlier, the transfer of benefits to more sport-specific tasks was not so obvious [30]. In relation to the issue of transfer, it is questionable if, from a functional point of view, training the ability to follow faster and faster objects really allows one to adequately distribute one's attention in space in order to follow

several players scattered on the field. On the other hand, regarding the study of the relationship between sports performance and visual tracking performance, it is questionable whether the objects' velocity used to observe differences between participants [1, 15, 16, 18, 19] was consistent with the velocity of objects to monitor in real-world. Depending on the study, the apparent velocity of moving objects was 16.8, 21.2 and 25.5 deg/s [19], or was 7.4, 9.9 and 12.4 deg/s [18] or was 5, 10 and 15 deg/s [1]. Jin and colleagues did not observe any tracking performance differences between basketball players and non-basketball players when objects moved at 5 deg/s [1], although this velocity seems reasonable given the constraints of the field since some of players to monitor can move at low speed or even be immobile. If the situational features used in the laboratory are not consistent with those of the real world, one may wonder about the generalization of the results to the field and their practical implications [28]. Speed is not the only situational feature that makes it difficult to track multiple objects in real-world scenarios. Beyond the speed factor itself, increasing the speed of the objects increases the spatial interference between the targets and the disruptors, which increases the difficulty of tracking [36]. Crowding can also be a situational feature that can complicate tracking in real-world scenarios [37].

To address these two concerns, more sport-specific visual tracking tasks could be considered. A first step in this direction would be to specify the behavior of the stimuli to the studied domain of expertise to increase the correspondence between the demands of the tracking task and the attentional demands of the field [28]. But to our knowledge, no study has yet investigated the relationship between sport performance and tracking performance using stimuli with sport-specific movements. Therefore, in this paper, a multiple players tracking task was proposed to investigate the relationship between sports practice and visual tracking performance in a virtual soccer environment. Two objectives arose from the above concerns. The primary objective was to study the interaction effect of sport practice and the specificity of virtual players movements on tracking performance. For this purpose, a virtual environment has been designed, in which virtual players moving on a soccer field had to be tracked by participants. Depending on the experimental conditions, virtual players moved either following real soccer game trajectories in a structured collective behavior (STRU) or following pseudo-random trajectories in an unstructured collective behavior (UNSTRU). Tracking performance was then assessed among participants divided into a group of soccer players (SOCC), a group of team sport players (TEAM) practicing a team sport other than soccer and a group of non-team sport participants (NoTEAM). As team sport players demonstrated higher visual tracking performance than non-team sport players in generic MOT tasks [1, 17–19], we expected to observe a main group effect such that SOCC and TEAM would outperform NoTEAM regardless of condition. Then, and of most interest, we expected to observe an interaction effect between group and condition, such that the difference in visual tracking performance between SOCC and the other two groups was greater in the STRU condition than in the UNSTRU condition. Because of their knowledge and experience, soccer players could have an advantage over other participants when the tracking task involved soccer-specific movements. TEAM group was added in the investigation to see if the soccer-specific movements used are beneficial only to soccer players and not to other team sports players. North and colleagues (2016) compared the accuracy of high-skill and less-skill soccer players to anticipate a game situation outcome when viewing video-film and point light display formats [25]. They observed that high-skilled soccer players made more accurate anticipations when watching a video-film or a point light display (without any discerning features) [25]. Thus, they suggested that the perception of structured collective behaviors alone was sufficient to anticipate the outcome of the situation [25]. Only manipulating virtual players' trajectory, and no other visual features, would be sufficient to improve the naturalism of the visual tracking task and to benefit the soccer

players [28]. The secondary objective of this study was to identify the situational features that have influenced the participants' performance in the proposed virtual players tracking task. From this perspective, the manipulation of situational characteristics was not possible with real scenarios without distorting them. If we substantially increased the velocity of the players to see how the velocity feature affected tracking performance, the real-world scenario would not have been so real. To preserve the naturalness of real-world scenarios, we conducted an original exploratory analysis with a clustering of virtual players to track, *i.e.* target-players, based on their successful tracking ratios and situational features identified as affecting the task. Because the target-players had their own situational features, we assumed that the probability of being correctly tracked would not be equal among them. Some targets should therefore be more difficult to track than others. The exploratory analysis aimed to describe the situational features associated with hard-to-track target-players. This analysis was applied to the STRU and UNSTRU conditions to see whether the hard-to-track target-players were described with a different set of situational features when virtual players were driven with real-world or pseudo-random trajectories.

## Materials and methods

### Participants

50 participants with normal or corrected-to-normal vision voluntarily completed this experiment (age: 24.6 ± 5.5 years). Based on their answers to a questionnaire about their sport practice, 16 participants (8 females, 8 males) were assigned to NoTEAM group, 16 participants to SOCC group (2 females, 14 males) and 18 participants (4 females, 14 males) to TEAM group (4 rugby players, 13 handball players and 1 basketball player). In SOCC, 6 participants competed in district level, 8 participants competed in regional level and 2 participants competed in national level. In TEAM, 2 participants competed in district level, 2 participants competed in regional level, and 14 participants competed in national level. All competition levels are relative to French championships. Prior to participation, all volunteers were informed about the purpose of the study, the information collected, their possibility to withdraw from the study at any time with deletion of their data and their consent to participate were collected through an online form. At the end of the form, participants agreed to participate by ticking the appropriate checkbox, entering their email address and clicking a submit button. They then received an email with a personal link to access the online application to participate in the study. The data collection procedure was validated by the local ethics committee of the University Rennes 2 (reference number: 2021–012).

### Data collection

Data collection took place during the COVID-19 pandemic, which involved severe constraints of social distancing and did not allow participants to access the laboratory experimental facilities. Data collection was therefore carried out via a web application. This online process did not ensure optimal standard laboratory conditions, but still allowed access to a motivated population that deliberately participated in the study. Online applications has already been used to assess perceptual performance of participants [38]. The virtual environment was generated with the Unity 3D engine and deployed in a WebGL version so that the application ran on the browser of participants in full screen mode.

## Experimental procedure

After agreeing to participate and filling an information questionnaire, participants conducted a training phase of 4 trials to become familiar with the environment and the proposed visual tracking task. Then, the evaluation protocol itself consisted of 30 randomized trials, including 15 trials for the STRU condition and 15 trials for the UNSTRU condition. Only the results of the evaluation phase are kept for data analysis. The number of trials (15 per condition) may seem small compared to other investigations on visual tracking expertise [17, 19], but it was intended to encourage participants to remain focused until the end of the experiment. In addition, inter-individual differences in visual tracking performance can appear even with a very low number of trials per condition [18]. The total time to complete the whole experiment from the information questionnaire to the training phase to the evaluation phase was estimated to be about 45 min. Participants were asked to use a computer (no smartphone or tablet) and to position their eyes at a distance from the screen equivalent to the width of their screen. In this way, the screen should have subtended approximately 53 degrees of visual angle for all participants.

## Visual tracking task

Participants observed the scene from a vantage point corresponding to that of a defender on a virtual soccer field (Fig 1), elevated to a height of 2.5 m to limit occlusions between virtual players. No virtual soccer ball was present in the virtual environment. Each trial consisted of three consecutive steps. a) Identification step: 4 target-players (2 blue and 2 red) to track were highlighted and pointed by an arrow. The participant launched the tracking step by pressing the space bar once he had correctly identified the 4 target-players. b) Tracking step: highlighting and arrows disappeared so that the target-players became visually identical to the other virtual players and, after a delay of 2 seconds, all virtual players started to move. Participants had to track the 4 target-players for 10 seconds, after which all virtual players stopped moving. c) Selection step: the participant had to click with the mouse on the target-players he thought he had successfully tracked. Unselecting a virtual player was possible by clicking a second time on it. Once his selection was complete, the participant had to press a validation button to end the trial.

In the proposed tracking task, we chose not to restrict the movement of the virtual players (including target-players) to the viewport displayed on the computer screen. In order to keep track of them, the participant could only rotate the vantage point around the vertical axis, *i.e.* to the left or right, using the corresponding directional arrow keys. No other modification of

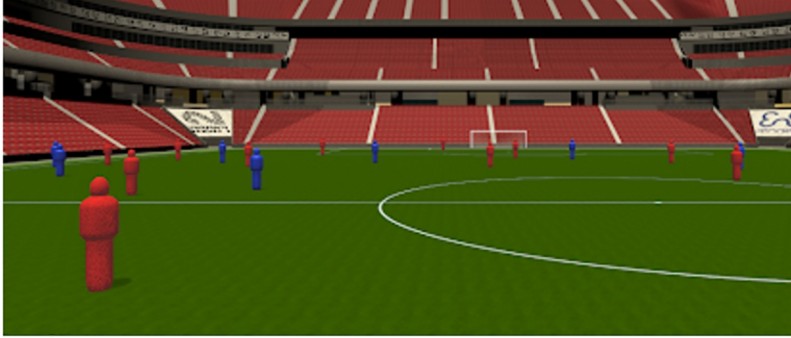

**Fig 1. Screen capture of the soccer virtual environment.** Virtual soccer players are presented as mannequins in blue for the defenders and in red for the attackers.

the vantage point was possible. However, the maximum angular distance between the target-players never exceeded the angular range covered by the viewport, *i.e.* 90˚. By rotating the vantage point appropriately, all the target-players could remain within the viewport at any time during each trial. Previous studies have shown that changes in vantage point affect participants' visual tracking performance [39, 40]. In the study by Huff and colleagues, participants' vantage point abruptly rotated around the scene during the visual tracking task [39]. In Thomas and Seiffert's study, participants rotated around the scene passively or actively [40]. In both studies, tracking performance was reduced when participants changed their vantage point [39, 40]. Because self-motion interferes with multiple object tracking [40], allowing the pivot of the vantage point probably affected participant tracking performance in this study as well. However, the difference here was that the participants did not rotate around the scene, but reorient themselves to track the players as they would on a real soccer field. They controlled the rotations themselves and did so to successfully complete the visual tracking task. On a real soccer field, players engage in active scanning behaviors, moving and rotating their bodies and heads, to gather information about their surroundings [41, 42]. Jordet and colleagues suggested that this active visual exploration to overcome spatial field constraints has a positive effect on soccer-specific performance [43]. This component was addressed in the study by Ehmann and colleagues [44]. Visual tracking performance was assessed in 360˚ MOT scenarios that allowed participants to reorient themselves during the visual tracking task to mimic the spatial constraints of the soccer field [44]. They then observed age-related changes and the effect of soccer performance on the ability to track multiple moving virtual players in a 360˚ environment [45]. Thus, we deliberately chose to add this rotation component in an attempt to get closer to the constraints of the field, *i.e.*, that the players to be monitored are moving all around and not just on a frontal plane.

## Selection of real game situations and generation of unstructured situations

15 situations have been extracted from a database of 7,500 situations recorded during real games (Data Source: STATS, copyright 2019 [46]). It was unclear from which actual soccer games the data were provided, but the data came from high-level championships, as this level of competition was the most likely to have a tracking system. In addition, we assumed that situations have still been familiar to district level soccer players, even if the situations come from high-level championships, because there is a common logic of play between the different levels of competition. The situations have been filtered according to their duration (> 10s), and according to some situational characteristics such as the average movements of both teams along the length of the field to exclude counter-attack situations, and along its width, to exclude corner-like situations. Remaining situations were then sorted according to the average speed of players, their average acceleration, their average dispersion, and the distance between the centroid of each team. Situations with the highest average speed, acceleration, or dispersion and the shortest inter-centroid distance were prioritized for visual inspection by one of the investigator, who ultimately selected 15 situations with multiple player crossings. For each of the 15 structured situations (STRU), a corresponding unstructured situation (UNSTRU) was computed as described in Fig 2. We assumed the displacement of virtual players in UNSTRU situations had no collective coordination or soccer game logic. The average traveled distance of virtual players in UNSTRU was approximately equal to that of the corresponding STRU situation.

## Data analysis

R software (version 4) was used for statistical processing of the data. Visual tracking performance was measured as the ratio of the number of correctly selected target-players to the total

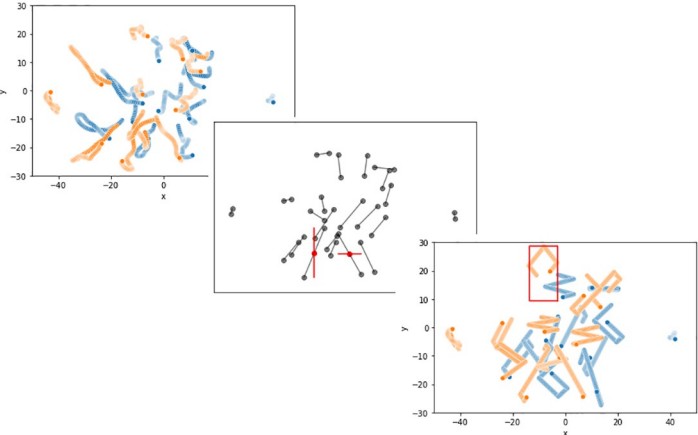

**Fig 2. Trajectories of virtual players over time in a STRU and UNSTRU situations.** In each plot, the axes represent the coordinates of the soccer field in meters with the origin at the center of the field. Ticks and units have been removed from the x and y axis of the central plot for clarity. Top Left) Original STRU situation: the virtual players followed real game trajectories. The orange trajectories correspond to the players of the team carrying the ball. The blue trajectories are those of the players of the opposing team. Center) From STRU to UNSTRU situation, virtual players trajectories were removed, only initial and final positions were maintained. The initial and final positions are gray dots. Each final position is connected to its initial position by a gray line, just for this illustration. The maximal distances on field length and field width (in red) that separate an initial position to the corresponding final position were used to dimension an imaginary box with an additional margin of 1m by side. This imaginary box was used to constrain virtual players' path. If an imaginary box exceeded the field limits, its dimensions were reduced to match the boundaries of the virtual soccer field. Bottom Right) Computed UNSTRU situation: A random starting direction was given to each virtual player which then moved in a straight path at constant speed and only changed direction when hitting a wall of its imaginary box (in red). The orange trajectories correspond to the players of the team carrying the ball. The blue trajectories are those of the players of the opposing team.

number of target-players to track (see S1 File). A mixed-model ANOVA was conducted to observe the within-subjects effect of experimental condition (STRU and UNSTRU), the between-subjects effect of sport practice (SOCC, TEAM and NoTEAM) and their interaction effect on visual tracking performance. The significance level was set to 0,05. Effect size (partial eta squared: $\eta_p^2$) was computed for each main effect. Pairwise T-Test with Bonferroni correction has been used for post-hoc analysis. Normality and homogeneity of variances assumptions were verified with Shapiro-Wilk test and Levene's Test respectively.

For the exploratory analysis, a Principal Component Analysis (PCA) and a Hierarchical Clustering on Principal Components (HCPC) has been conducted for both experimental conditions (STRU and UNSTRU) using the FactoMineR package [47]. It aimed to cluster target-players with respect to their successful tracking ratio and situational features. Four situational features known to challenge the tracking task were investigated in this analysis, *i.e.* the speed of target-players [7, 48], the spatial dispersion of target-players [49] and the density of disruptors near target-players [8, 37]. As tracking demands increase with disruptors that match the color of the target [50], the density feature was investigated both by considering disruptors of any color and by considering only disruptors of the same color as the target-player. However, a target-player may also be lost because of the demands of the other three target-players involved in the given situation. A total of 8 situational features were computed for each target-players, *i.e.* 4 individual features were related to the target-player itself and 4 group features were related to the three other target-players of same situation. To complete the description of each target-player, its successful tracking ratio was computed for the 3 groups of participants. Therefore, each target-player was described by 11 centered reduced variables (see Table 1 for the

**Table 1. Description of investigated features in the exploratory analysis.**

| Acronym | Variable | Description |
|---------|----------|-------------|
| *Ratio.SOCC* | Successful tracking ratio for SOCC | Ratio of being successfully tracked by each group of participants |
| *Ratio.TEAM* | Successful tracking ratio for TEAM | |
| *Ratio. NoTEAM* | Successful tracking ratio for NoTEAM | |
| *Ind.Speed* | Individual speed | Average angular speed relative to the participant vantage point |
| *Ind.Density* | Individual density of disruptors | Frequency of disruptors within 3˚ of the target-player [8] |
| *Ind. DensityCol* | Individual density of disruptors of same color | Frequency of disruptors of same color within 3˚ from the target-player |
| *Ind. DistTarget* | Individual distance to closest target-player | Averaged minimum distance to nearest target-player |
| *Grp.Speed* | Group speed | Average angular speed of the three other target-players |
| *Grp.Density* | Group density of disruptors | Average frequency of disruptors within 3˚ of the other target-players [8] |
| *Grp. DensityCol* | Group density of disruptors of same color | Average frequency of disruptors of same color within 3˚ of the other target-players |
| *Grp.Disp* | Group dispersion | Average standard deviation of the angle of the three other target-players relative to the participant vantage point |

variables' description). The coordinates of the vantage point of the participants and the coordinates of each virtual player with respect to the field were used to compute for each virtual player, an angle between the longitudinal axis of the field and the line connecting the virtual player to the vantage point. These angles were used to compute the situational features listed in Table 1. As participants had exactly the same vantage point in the virtual world, the perceived situational features were the same for all participants. For both experimental conditions, a data table of 60 rows (4 target-players in each of the 15 situations) by 11 columns (3 tracking ratio variables, 4 individual features and 4 group features) was given as input to the exploratory analysis (see S2 File for STRU condition and S3 File for UNSTRU condition). The PCA pre-processing step was performed to reduce the dimensionality of the multivariate dataset and the number of principal components (PCs) to consider was chosen to explain 70% of the total variance. The retained components were used in the HCPC step to cluster target-players based on their similarities. The number of clusters returned by the HCPC was automatically set to maximise inter-cluster differences and minimize intra-cluster differences. Association tests conducted in the HCPC procedure indicate whether the mean values of investigated variables of each cluster are significantly different from the mean values of all 60 target-players for the two experimental conditions of virtual player movements.

## Results

### Visual tracking performance

This study declares no missing data. The mean individual tracking performances are shown in Fig 3 and the mean tracking performance by group is presented in Table 2.

The mixed-model ANOVA revealed a significant effect of the experimental condition (F$(1,47)$ = 33.718, p<0.001, $\eta_p^2 = 0.418$), a significant effect of group (F$(2,47)$ = 3.537, p = 0.037, $\eta_p^2 = 0.131$), but no interaction effect between these two factors (F$(2,47)$ = 0.844, p = 0.437, $\eta_p^2 = 0.035$). Pairwise T-Test comparisons revealed higher tracking performance for SOCC

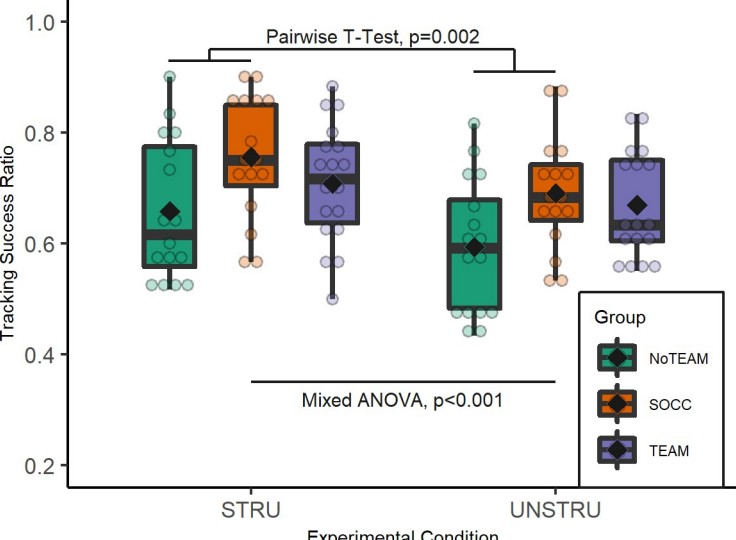

**Fig 3. Tracking success ratio by group in both experimental conditions.** Boxplots show distribution of subjects'
performance (fade dots) and ♦ represent group mean.

than NoTEAM (p = 0.002), but no difference between TEAM and NoTEAM (p = 0.071) or
between SOCC and TEAM (p = 0.63).

## Exploratory analysis for target-players clustering

The clustering procedure was conducted for both experimental conditions. After a PCA step,
the HCPC function of FactoMineR package was applied to the retained PCs to cluster target-
players [47]. The HCPC function also returned results about association tests indicating
whether the mean variable of a cluster is significantly different than the mean variable of all
target-players [47] (see S4 File for HCPC on STRU experimental condition and S5 File for
HCPC on UNSTRU experimental condition). This exploration analysis focused on the most
significantly situational features associated to clusters of hard-to-tracked target-players for
both experimental conditions.

**STRU condition.** The first four PCs from the PCA, with respective variances of 27.2%,
20.6%, 16.3% and 8.7%, were retained to express a cumulative variance of 72.9%. The biplot in
Fig 4A displayed the 11 inspected variables and the 60 individual target-players in the plane
formed by PC1 and PC2. The 3 successful tracking ratios were fairly well represented
($cos^2 > 0.6$) on this plane which suggested hard-to-track target-players seemed the same for all
3 groups of participants. Based on the distance between target-players in the 4 retained PCs,
the HCPC algorithm grouped the 60 target-players into three distinct clusters (Fig 4A). For the
sake of brevity, only the first two components are shown in Fig 4A. The plot may be misleading
due to the significant overlap of the three target player groups, but as a reminder, the third and

**Table 2. Description of the mean (standard deviation) tracking performance by group and condition.**

|  | SOCC (n = 16) | TEAM (n = 18) | NoTEAM (n = 16) |
|---|---|---|---|
| **STRU** | 0.756 (±0.110) | 0.708 (±0.107) | 0.658 (±0.128) |
| **UNSTRU** | 0.691 (±0.103) | 0.669 (±0.093) | 0.594 (±0.122) |

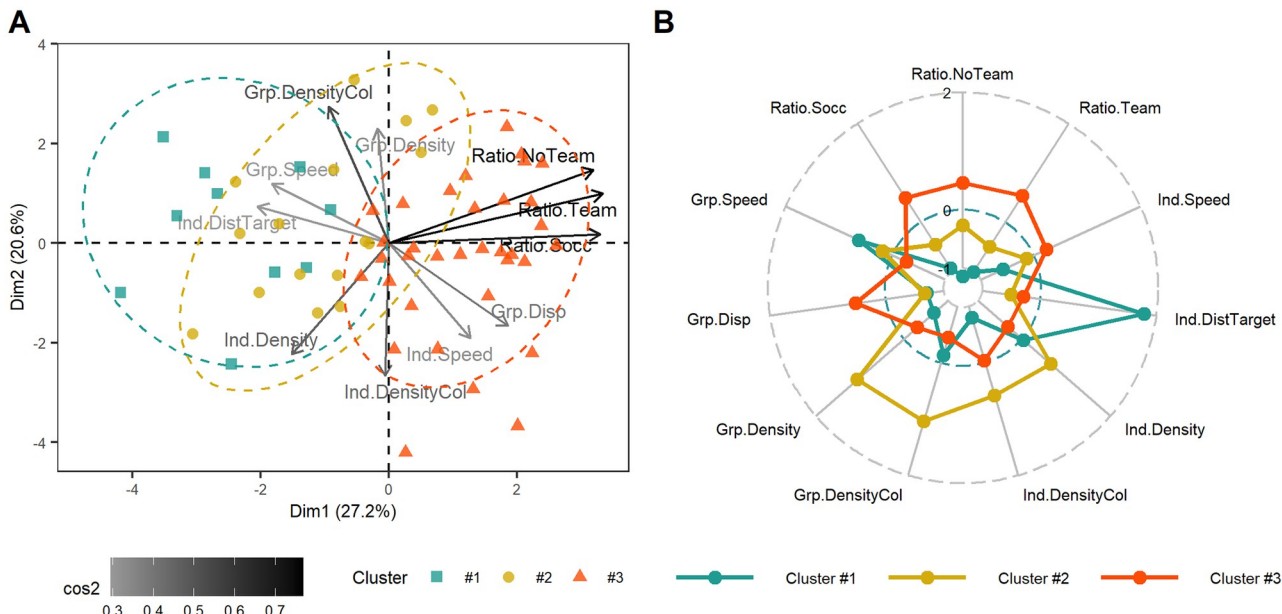

**Fig 4. Clustering of target-players in STRU condition.** A) PCA Biplot with the investigated variables and individual target-players projected on the first two dimensions returned by the PCA. The color gradient expresses the amount of variance conserved on this plane for each variable. The individual target-players' shape and color are given with respect to their respective cluster computed with the HCPC algorithm. B) Radar plot with centered-reduced variables taken by each cluster of target-players returned by HCPC algorithm. The more central the plot of the variable, the lower its value and conversely. The middle circle line stands for the mean value taken by each variable considering all target-players. The more the value of a cluster is far from the middle circle line, the more the target-players of this cluster are different from the others.

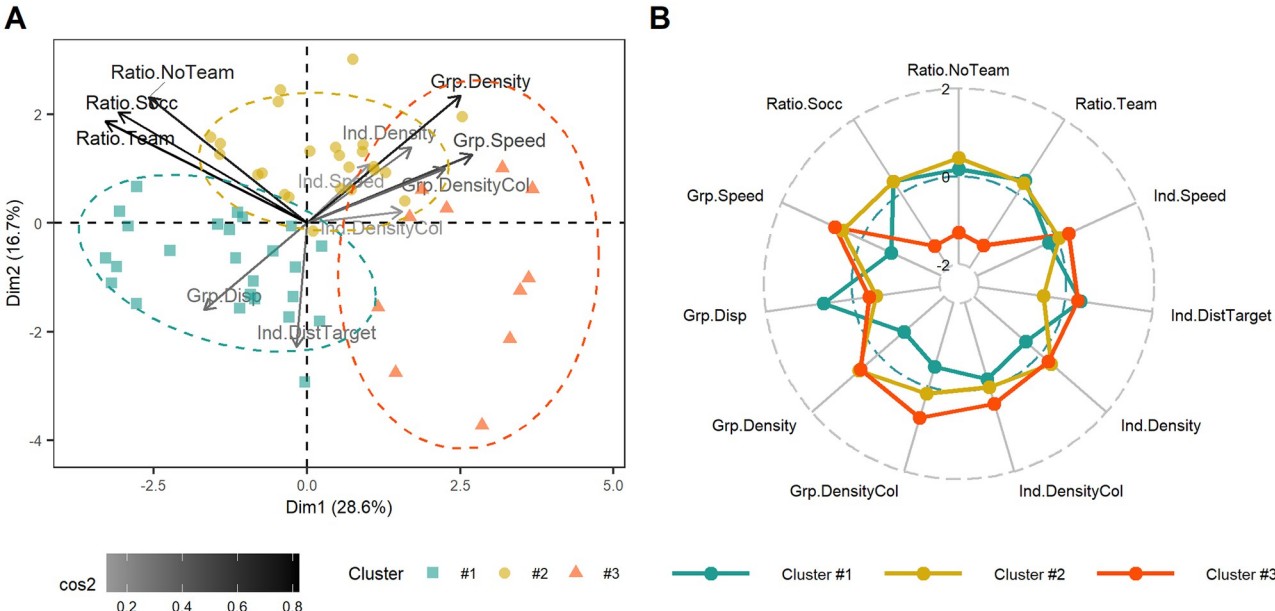

**Fig 5. Exploratory analysis for target-players of UNSTRU condition.** A) PCA Biplot with the investigated variables and individual target-players projected on the first two dimensions returned by the PCA. The color gradient expresses the amount of variance conserved on this plane for each variable. The individual target-players' shape and color are given with respect to their respective cluster computed with the HCPC algorithm. B) Radar plot with centered-reduced variables taken by each cluster of target-players returned by HCPC algorithm. The more central the plot of the variable, the lower its value and conversely. The middle circle line stands for the mean value taken by each variable considering all target-players. The more the value of a cluster is far from the middle circle line, the more the target-players of this cluster are different from the others.

fourth PCs were also retained for clustering. The mean values of the 11 variables are presented in different colors for each of the 3 clusters of target-players in the radar plot in Fig 4B.

Cluster #1 consisted of 10 target-players that were associated with significantly lower Ratio. NoTeam, Ratio.Team, Ratio.Socc, Ind.DensityCol, Grp.Disp, Grp.Density and significantly higher Ind.DistTarget, Grp.Speed than the global average (S4 File). Cluster #2 consisted of 16 target-players that were associated with significantly lower Grp.Disp, Ind.DistTarget, Ratio. Team, Ratio.Socc and with significantly higher Grp.Density, Grp.DensityCol, Ind.Density, Ind.DensityCol than the global average (S4 File). Cluster #3 consisted of 34 target-players that were associated with significantly lower Grp.DensityCol, Ind.Density, Grp.Density, Ind.DistTarget, Grp.Speed and with significantly higher Ratio.Team, Grp.Disp, Ratio.Socc, Ratio. NoTeam, Ind.Speed than the global average (S4 File).

**UNSTRU condition.** The first four PCs from the PCA, with respective variances of 28.6%, 16.7%, 15.0% and 10.3%, were retained to express a cumulative variance of 70.6%. The biplot in Fig 5A shows the 11 inspected variables and the 60 individual target-players in the plane formed by PC1 and PC2. The 3 successful tracking ratios were fairly well represented on the plane ($cos^2 > 0.6$) which also suggested hard-to-track target-players seemed the same for all 3 groups of participants. Other situational features that seemed fairly well represented in PC1 and PC2 were Grp.Density and Grp.Speed ($cos^2 > 0.6$). As indicated by the arrows, the ability of a target-player to be tracked might be anti-correlated to the density of nearby disruptors and the speed of the three other target-players present in same situation. Based on the distance between target-players in the 4 retained PCs, the HCPC algorithm grouped the 60 target-players into three distinct clusters (Fig 5A). The value of the 11 variables for the 3 clusters of target-players are presented in the radar plot in Fig 5B.

Cluster #1 consisted of 25 target-players that were associated with significantly lower Grp. Density, Grp.Disp, Grp.Speed, Grp.DensityCol, Ind.Density and significantly higher Ratio. Team, Ind.DistTarget, Ratio.Socc than the global average (S5 File). Cluster #2 consisted of 24 target-players that were associated with significantly lower Grp.Disp, Ind.DistTarget and higher Grp.Density, Grp.Speed, Ratio.NoTeam, Ind.Density and Ratio.Socc than the global average (S5 File). Cluster #3 consisted of 11 target-players that were associated with significantly lower Ratio.Socc, Ratio.Team, Ratio.NoTeam and significantly higher Grp.DensityCol, Grp.Speed than the global average (S5 File).

## Discussion

### Visual tracking performance

The average tracking performance of participants was significantly higher in STRU condition than in UNSTRU condition. Although the results of this study did not provide a definitive explanation for this difference, it is likely that virtual players movements in UNSTRU situations were more unpredictable and therefore more difficult to track (see [51] for a manipulation of entropy in object movement). In UNSTRU situations, the virtual players could abruptly change direction without changing speed and it was not possible for participants to predict the timing and direction of the turn. Whereas in STRU situations, since the trajectories are extracted from real soccer fields, the virtual players decelerated before changing direction.

A significant group effect was observed in this study where only SOCC significantly outperformed NoTEAM regardless of the experimental condition. This result was partially consistent with the literature that reported higher tracking ability in team sport players than in non-team sport players when facing stimuli with non-specific movements [1, 17, 18]. A fairly wide dispersion of intra-group values was observed, suggesting that factors other than simple sport practice probably affected participants' performance, such as field position [19], level of

competition [17] or frequency of video game use [11]. Regarding the level of competition, the small number of participants who practiced team sport (16 soccer players, 18 other team sport players) did not allow for a comparison of balanced groups of participants with different competition levels (8 in district, 10 in regional and 16 in national level). It was therefore not possible to state whether the group effect was solely related to the mere practice of team sport or whether it was affected by any competition level effect [17, 21]. The non-difference in tracking performance between SOCC and TEAM may be partially explained by the practice of team sports that may present similar tracking demands than soccer [52]. Rugby, handball and basketball are also invasion games that involve many interactions between teammates and opponents resulting in collective behaviors that could look similar to those in soccer. This would allow for the transfer of tracking skills between team sport activities [52].

Soccer players do not simply monitor the individual movements of their teammates and opponents on the field, but look for structured collective movements emerging from these individual movements to help them anticipate the evolution of a game situation [25]. Thanks to their specific knowledge, the tracking demands could have been reduced for soccer players when the stimuli were directed with real game trajectories [26]. Contrary to our expectation, visual tracking performance difference between soccer players and other participants was not greater when faced with stimuli involving soccer-specific movements compared to stimuli involving pseudo-random movements. This result suggests that directing virtual players with real game trajectories was certainly not sufficient to recreate the game conditions under which players have to track teammates and opponents on the field.

It seems likely that people who regularly watch soccer games would also exhibit enhanced visual tracking abilities in situations with soccer-specific trajectories, even if they do not necessarily play soccer themselves. In this regard, participants were asked to indicate how often they watched soccer games on a 5-point Likert scale (Never—Once a year—Once a month—Once a week—Several times a week) in the online form prior to performing the visual tracking task. The results of a two-way mixed ANOVA revealed a significant effect of condition ($F(1,45) =$ 28.170, $p<0.001$) but no effect of viewing frequency ($F(4,45) = 0.653$, $p = 0.628$) and no interaction effect between these two factors ($F(4,45) = 0.647$, $p = 0.631$) on visual tracking ratio. It appears that participants with higher viewing frequency of soccer games did not show better visual tracking performance, whether faced with situations with soccer-specific trajectories or not. In addition, the results of a one-way ANOVA revealed a significant main effect of the group on viewing frequency ($F(2) = 18.501$, $p<0.001$). In pairwise comparisons, a higher viewing frequency was observed in SOCC than in TEAM ($p<0.001$), a higher viewing frequency was observed in SOCC than in NoTEAM ($p<0.001$), but no differences were observed between TEAM and NoTEAM ($p = 1.000$). Thus, the groups did not differ equally in viewing frequency of soccer games and visual tracking performance. Overall, the difference in visual tracking performance between participants in situations with soccer-specific trajectories cannot be explained solely by whether or not they regularly watch soccer games. What is important seems to rely more on the viewing perspective than on the viewing frequency. Attentional demands in the field differ depending on the playing position. Differences in visual tracking performance have been observed between frontcourt and backcourt basketball players [16], and between frontward and backwards rugby players [19]. The defender's viewpoint was chosen in this study because most teammates and opponents can be seen within a maximum visual angle of 180°, which increased the crowding of the field of view by the players. With a more advanced viewpoint on the field, the players to be monitored would have been distributed 360° around the observer. A similar defender viewpoint was used in the study by Roca and colleagues (2013) to investigate participants' anticipation when viewing videos of soccer game situations [24]. It can be assumed that defenders would perform better than strikers in

the current investigation. It would be interesting to see if playing position would have also influenced tracking performance in this study, but the sample size of soccer players group did not allow for this statistical analysis (3 strikers, 6 midfielders, 6 defenders, and 1 participant did not fill in his/her playing position).

The ball and the ball carrier are important visual cues that soccer players must focus on to anticipate the outcome of a game situation [24]. North and colleagues (2017) observed that soccer players were more accurate in recognizing familiar soccer situations presented in point light displays when the ball was present than when it was not [53]. In situations with soccer-specific trajectories, the presence of the ball would probably have helped to perceive structured collective behaviors and thus the visual tracking of several players. However, it was difficult to estimate how the presence of the ball would have influenced participants' visual tracking in situations with pseudo-random trajectories. Thus, the ball was removed from the entire experiment because its presence would surely have affected participants' visual tracking performance differently between the two experimental conditions. The influence of ball presence on the visual tracking in situations with soccer-specific trajectories should be studied in future research.

Some limits regarding the experimental procedure have to be highlighted. Despite the advantage of allowing data to be collected online [38], the use of a web-based application did not allow control of the conditions under which participants performed the task, particularly with regard to compliance with the instructions relating to the distance between the eye and the screen. This instruction was made to ensure standardization of viewing conditions, as visual tracking performance can be influenced by the tracking area size and the speed of targets [54]. For future online investigations, validated methods are now available and should be considered for controlling viewing distance and stimulus size [55, 56]. Regarding the stimulus size issue in the current investigation, the task could only be performed in full screen mode. The width and height of the browser viewport of participants have been recorded to control the influence of screen dimension on visual tracking performance. The height of the windows varied from 578 to 1137 pixels and the width of windows varied from 960 to 1920 pixels. Pearson's correlation between mean tracking performance and screen height was 0.141 (p = 0.425). Pearson's correlation between mean tracking performance and screen width was -0.360 (p = 0.306). Therefore, participants' visual tracking performance is assumed not to have been influenced by the size of the participants' computer screen. However, the participants' browser frame rate was not recorded. Latency may have influenced the speed of objects and disrupted the fluidity of movement during the task. Although participants did not complain about latency issues, it can only be assumed without guarantee that few or no latency occurred during the experiment. During pre-tests with different computers, browsers and networks, no latency has been experienced. The data required to execute a trial was loaded into the browser's cache before the trial began, which avoided network latency as the virtual players moved. Another limiting factor is that all participants certainly did not comply with the instruction to select only those target-players whom they were confident enough to have successfully completed the tracking. However, we assume that this bias induced by the possibility of guessing the target-players during selection was minor since the probability of guessing the correct targets in the middle of 17 distractors is low by pure chance. Moreover, finding a real target-player after guessing among some nearby virtual players still reflected a good level of visual tracking ability.

Other limits can be noted regarding the use of 2D stimuli that were constrained by the frame of a computer. First, depth information, which is important for tracking multiple objects in 3D space [57], was not available on the 2D screen. Second, the interaction medium that allowed exploration of the environment (i.e., viewpoint pivot with keyboard keys without

translation) was poorly representative of the possibilities of visual exploration in the field [58]. This did not allow soccer players to engage more elaborated information gathering behaviors like body position adjustments [44]. Abrupt changes in vantage point and rotation around the scene during a visual tracking task were observed to have a negative effect on visual tracking performance [39, 40]. The rotation of the vantage point generated a very short period during which it was not possible to collect information, as during an ocular saccade [37]. It would also likely have affected the visual tracking performance of the participants in this study but, like the eye saccade, it was a behavior controlled by the participant to adequately explore the virtual environment. We strongly believe that it is necessary to retain this component of vantage point adjustment in the investigation of tracking performance of soccer players, as it is an inherent behavior of soccer practice. However, it seems preferable to use more natural means of exploration [58]. To address these limitations, we encourage next investigations to use immersive virtual reality technologies. This would allow participants to access depth information [59] and adopt more natural visual exploration behaviors [60] to track virtual players with more faithful attentional constrains as they move all around and not only in front of them [44]. Furthermore, postural and motor control affect attention allocation [61, 62]. Virtual reality facilities would allow participants to perform the tracking task while standing and even moving as a player would on the field. We assume that most participants completed the proposed tracking task in a seated posture, which did not represent the actual motor constraints of a soccer game and played a role in the allocation of attentional resources.

## Exploratory analysis for target-players clustering

The aim of this HCPC-based approach was to group target-players according to their ability to be successfully tracked and their situational features. For both experimental conditions, the 60 target-players were grouped into 3 clusters after dimensions reduction by PCA. This procedure allowed to describe the hard-to-track target-players in both experimental conditions (virtual players following soccer-specific or pseudo-random trajectories), while observing the associated situational features variations. Hard-to-track target-players were identified when the successful tracking ratio of a cluster was below the global average. Target-players in clusters #1 and #2 were considered as hard-to-track in STRU condition (Fig 4B) and target-players in cluster #3 were considered as hard-to-track in UNSTRU condition (Fig 5B).

Regarding the STRU condition, having two distinct clusters of hard-to-track target-players suggests that tracking performance was not affected by a single situational feature but rather depended on specific combinations of them. On the one hand, hard-to-track target-players from cluster #1 were likely to be lost because they were far from the other three target-players (high Ind.DistTarget) which were grouped together (low Grp.Disp) and which moved rather fast (high Grp.Speed). These hard-to-track target-players illustrated the difficulty to gather information on dispersed field areas, especially when one area is a major source of interest. Participants probably did not divide their attention equally among the four target-players, but rather focused on the three grouped target-players that moved in a coordinated manner in the same area. The grouping strategy likely ensured the success of the tracking on at least three target-players [63, 64]. Despite the ability to gather information on spread areas seemed to have a positive effect on soccer-specific performance on the field [43], the soccer players had the same difficulties as the other participants on these hard-to-track targets-players. On the other hand, soccer and other team sports players likely missed the target-players in cluster #2 due to visual interference induced by the density of the disruptors surrounding each of the target-players (high Grp.Density, Grp.DensityCol, Ind.Density and Ind.DensityCol). These results are consistent with the literature, which has observed a negative influence of density on visual tracking

performance [8, 37]. These hard-to-track targets-players reflected the difficulty in disentangling information in very dense situations where many players of both teams were located in the same field area.

Regarding the UNSTRU condition, the hard-to-track target-players in cluster #3were characterized by a very high density of same-colored disruptors around the other target-players (high Grp.DensityCol) involved in the same situation that moved fairly fast (high Grp.Speed). In contrast, the density of disruptors of any color appeared to be equal to that of cluster #2 target-players, which were fairly well tracked (Fig 5B). This supports the idea that color similarity with disruptors could imply visual interferences complicating the tracking of target-players [50]. Tracking demands associated to the other target-players crowding and speed may have been important enough to affect the tracking success of target-players in cluster #3 [48]. These hard-to-track target-players were in fact missed not because of their own situational characteristics but because of those of other target-players. However, since the density and speed features were not controlled here, it is not possible to conclude whether the speed factor increased tracking difficulty by itself or because faster moving target-players induced more visual interference [7, 36]. Thus, the only statement that can be made is that among all 60 target-players to be tracked in the UNSTRU condition, 11 target-players were regularly lost or confused in situation where movements were faster and with higher density of same color disruptors than average.

In summary, when comparing results across experimental conditions, the situational features associated with hard-to-track players differed. In STRU condition, isolated target-players (cluster #1) and target-players in very cluttered space (cluster #2) were more likely to be lost, whereas in UNSTRU condition, target-players had more chance to be lost when the others to monitor were moving fast and were crowded by same color disruptors. This result suggests that with real scenarios, the space distribution of virtual players had a stronger influence on visual tracking performance than with pseudo-random scenarios. Furthermore, as the initial and final locations of the virtual players were the same between a STRU situation and its associated UNSTRU situation, the only difference in the situational features of target-players arose from the distribution of the virtual players in space during the movement phase. This generated different clusters of target-players. On a soccer field, the players are not uniformly distributed on the field, but the density varies in different areas, depending on tactical objectives with respect to the location of the ball. This concern has already been investigated in the study by Vater and colleagues [37]. The authors observed that targets grouped in cluttered spaces attracted participants' gaze to reduce the negative effect of crowding [37]. Participants' gaze activity has not been recorded in this study, but they might also have allocated their foveal attention to grouped target-players in crowded spaces. In STRU condition, the influence of speed was not as clear. On the one hand, hard-to-track target-players in cluster #1 presented a significantly higher than global average Grp.Speed suggesting that speed feature may have had a negative effect on the visual tracking [48]. On the other hand, easy-to-track target-players (cluster #3) were target-players with high speed (Fig 4B). Conversely in the UNSTRU condition, the speed features have a more obvious negative effect on visual tracking performance since the hard-to-track target-players were described with the highest Grp.Speed and Ind. Speed (Fig 5B). Therefore, in contrast to the pseudo-random movements, the visual tracking performance with real-world movements have been more affected by features related to the spatial distribution of virtual players than by features related to speed. In addition, unlike in the STRU condition, no clusters clearly illustrated easy-to-track target-players in the UNSTRU condition. Apart from the 11 target-players in cluster #3, the remaining 49 had an equal probability of being successfully tracked. This suggests that real-world trajectories invited participants to focus more on certain target-players and leave others. These differences highlighted

the need to consider stimuli with real life movements to understand how a situational feature may influence the tracking of multiple targets in real life. For future work on this issue, we recommend to not alter the movements of virtual players by individually manipulating each situational feature, but rather to use the situational features associated with hard-to-track target-players as selection factors for real-world game situations. In concrete terms, the game situation should be selected based on the density of players in a given area of the field and target-players should be selected based on the clutter in their neighborhood. This would allow the study of the influence of the spatial dispersion of players on the visual tracking task when facing real world movements. In addition, this study reported an average speed of real-world target-players of 3.3 ± 1.6˚/s, which was lower than the range of speed studied in other sports performance studies [1, 18, 19]. Tracking players who run at a high pace might be more difficult than tracking players who walk slowly [48], but we encourage future research on the relationship between sport performance and visual tracking performance to also rely on spatial features and not just the speed feature. Although visual tracking training improved the ability to track multiple fast objects [15, 31, 33, 35], transfer to a sport-specific task may not have occurred because of the lack of correspondence of field stimuli [28]. Instead of training the ability to track objects in a small space (2D screen) that move faster than in the real world, it would be interesting to train the ability to track objects in a large [44] and cluttered [37] space, to see if higher transfer benefits to a sport-specific task would occur.

However, some limitations have to be reported on these results. First, only a non-exhaustive set of situational features was investigated in this study. Other features such as entropy [51] could be added to refine the characterization of clusters. In addition, features were expressed in terms of viewing angle and not in terms of field plane coordinates, which did not account for the absolute distance between a virtual player and the participant's vantage point. The perceived size of a target-player decreased with its distance to the observer's location, potentially increasing the difficulty of the tracking task [65]. In future investigations, it would make sense to add scene coordinate features in order to better appreciate the situational features of players moving on the soccer field. Finally, the exploratory analysis focused only on features that were averaged over the 10 seconds of tracking. The resulting values may give weak indications of the difficulty criteria if a target-player has been lost since the very beginning of the trial. Time series analysis would be of great value in identifying the moment when a target-player has been lost and to focus the analysis before that time, especially when dealing with very dynamic situations with many speed changes [36].

## Conclusion

The present web-based study compared participants' performance on a tracking task as a function of their sport practice when facing stimuli with and without soccer-specific movements. The overall analysis did not support the expectation that the difference in tracking performance between soccer players and other participants would be greater when faced with stimuli directed by soccer-specific trajectories. However, a complementary analysis has been made to determine which situational features affected the tracking ability of participants. An original exploratory analysis using the HCPC algorithm has then been used to identify different clusters of target-players based on their situational features and successful tracking ratios between STRU and UNSTRU conditions, providing a first understanding of what can be difficult in tracking players on a soccer field. To go further, next investigations might consider studying more stimuli features but above all using Virtual Reality to propose a more ecological experimental environment.

## Supporting information

**S1 File. The mean ratio of successful visual tracking by participants in both experimental conditions.**
(CSV)

**S2 File. The raw dataset (before the data is centered reduced) used for the clustering of target-players in STRU condition.** Each row represents an individual target-player and each column represents a situational feature (columns 2–9) or a visual tracking ratio (columns 10–12).
(CSV)

**S3 File. The raw dataset (before the data is centered reduced) used for the clustering of target-players in UNSTRU condition.**
(CSV)

**S4 File. The description of the clusters by the variables in STRU experimental condition.** The v.test is a statistical indicator of the difference between the cluster mean and the overall mean for the corresponding variable. the associated p-value indicates whether the cluster mean is significantly different from the overall mean.
(CSV)

**S5 File. The description of the clusters by the variables in UNSTRU experimental condition.**
(CSV)

**S1 Video. Video capture of a STRU situation and its associated UNSTRU situation.**
(MP4)

## Author Contributions

**Conceptualization:** Alexandre Vu, Anthony Sorel, Charles Faure, Richard Kulpa.

**Formal analysis:** Alexandre Vu, Anthony Sorel, Richard Kulpa.

**Funding acquisition:** Benoit Bideau.

**Investigation:** Alexandre Vu, Charles Faure, Antoine Aurousseau.

**Methodology:** Alexandre Vu, Charles Faure, Antoine Aurousseau.

**Project administration:** Anthony Sorel, Benoit Bideau, Richard Kulpa.

**Supervision:** Anthony Sorel, Benoit Bideau, Richard Kulpa.

**Validation:** Anthony Sorel, Benoit Bideau, Richard Kulpa.

**Writing – original draft:** Alexandre Vu, Anthony Sorel, Richard Kulpa.

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
