## [Decision Letter · Decision Letter 0]

4 Feb 2022

PONE-D-21-39830Visual tracking assessment in a soccer-specific virtual environment: a web-based studyPLOS ONE

Dear Dr. VU,

Thank you for submitting your manuscript to PLOS ONE. After careful consideration, we feel that it has merit but does not fully meet PLOS ONE’s publication criteria as it currently stands. Therefore, we invite you to submit a revised version of the manuscript that addresses the points raised during the review process.

Two expert reviewers have assessed your work. Both reviewers find the topic of your study to be interesting, yet they also raise some critical issues. The reviewers ask for missing methodological details which are important to include and discuss. Both reviewers also raise important questions regarding the methodological limitations of the setup (the effect of allowing viewpoint changes) and the interpretation of your analyses (particularly the exploratory PCA and HCPC analyses). At this point it is unclear to me whether it is possible to draw solid conclusions from your work, and thus whether your paper can ultimately be published. Nevertheless, I would like to give you the chance to defend or modify your methodological choices. If you decide to revise and resubmit your potentially interesting study, I suggest you pay careful attention to replying convincingly to all reviewer comments. Additionally, I point out that the raw data underlying your results need to be made fully available. You may include these data in the supplementary files, or better still you could upload your raw data and analysis scripts to a public data repository (e.g. Zenodo) and include the doi linking to your data in your revised manuscript.

We look forward to receiving your revised manuscript.

Kind regards,

Guido Maiello

Academic Editor

PLOS ONE

Journal Requirements:

a) Did participants provide their written or verbal informed consent to participate in this study?

 [This study was partially funded by the ANR within the framework of the PIA EUR DIGISPORT project (ANR-18-EURE-0022) and by the Region Bretagne.]

4. Please ensure that you include a title page within your main document. You should list all authors and all affiliations as per our author instructions and clearly indicate the corresponding author.

Reviewers' comments:

Reviewer's Responses to Questions

**Comments to the Author**

1. Is the manuscript technically sound, and do the data support the conclusions?

Reviewer #1: Yes

Reviewer #2: No

2. Has the statistical analysis been performed appropriately and rigorously? 

Reviewer #1: Yes

Reviewer #2: Yes

3. Have the authors made all data underlying the findings in their manuscript fully available?

Reviewer #1: Yes

Reviewer #2: No

4. Is the manuscript presented in an intelligible fashion and written in standard English?

Reviewer #1: Yes

Reviewer #2: Yes

5. Review Comments to the Author

Reviewer #1: The authors studied how an observer visually tracks moving elements in a sport environment. They reduce a bias present in MOT protocols by simulating the movements of players on a field thanks to player movements extracted from a database of real football matches.

The authors compare the tracking performances of football players, athletes who do not play football but another team-based sport, and non-sport-practising individuals. They also have participants track scenes where virtual players follow real game trajectories, and also when the virtual players follow non-natural movements. Using ANOVA and HCPC analyses, the authors wished to analyse what experimental factors influenced tracking in the virtual scenes.

The presentation of the study, its details and findings were well presented and easy to understand.

* Method

Is it possible that the fact that virtual players are showing no discerning features (beyond their team's colour) may have made the task harder than it would be in a real-case scenario? Does the choice of making all players identical reflect a belief of the authors that visual tracking in soccer is a process requiring very few visual features?

I ask because the authors state their interest in approaching more natural conditions several times in the manuscript, but there was no discussion of realism related to visual features of the virtual players.

Was it really necessary to give participants the possibility to pan the camera view?

The authors point to this as a limit in the discussion section because some participants may have had issues manipulating the camera, on top of tracking elements in the scene. This is potential a bias that could have been easily avoided by restricting the movements of the virtual player to be within the camera viewport.

* Results

I understand that using a method such as HCPC allows for the study of many variables at once and to identify and characterise varied behaviours. But this appears rather post-hoc and nebulous to me in the interpretations made from the resulting data. It is complicated to measure and attribute the effects amongst the variables related to each group.

Additionally, although fig. 5.A shows fairly separable clusters of target-players, that is not the case for fig. 4.A.

I realise that the alternative would be to vary each variable independently and measure their resulting effect, therefore multiplicating the number of experiments needed to run.

But considering the fact that the authors present HCPC as an exploratory tool, I would at least expect to read in the discussion section about what they would do in future work to study more precisely the specific behaviours the HCPC analysis highlighted.

* typos

p.2 "players movements" -> "player movements"

p.3 "allow participants to access to the" -> "allow participants to access the"

p.6 "not difference" -> "no difference"

Reviewer #2: This paper investigates whether playing a team sport effects tracking performance in a visual tracking task using movement trajectories taken from soccer matches (i.e., structured). The authors compared the tracking performance of soccer players, other team sport players, and non-team sports players. They also compared performance on structured and unstructured movement trajectories. The authors did not find that soccer players displayed better tracking performance when facing the structured movement trajectories taken from real soccer games compared with the other two groups. Since the results did not fit with the authors’ hypothesis, they suggest an immersive design that captures the soccer field more closely should be used in future research.

The manuscript fits nicely within the scope of PLOS ONE and addresses an interesting question. However, there are several methodological limitations (many of which the authors outline themselves in the discussion) that make me wonder about the quality of the data and makes interpretation of the results somewhat difficult.

Major Comments

1. The introduction is clear and includes some relevant literature. However, I do think it could be more focussed towards the specific questions the authors are interested in addressing.

a. Expertise effects could be discussed in more detail. Quite some relevant research has been done into video game players (e.g., Dye & Bavelier, 2010; Green & Bavelier, 2006; Sekuler et al., 2008).

b. I would appreciate some discussion on why these conflicting results regarding expertise differences have been found. For example, why does expertise in some sports seem to lead to better MOT performance (10 – 13) but not in others (15)?

c. Following on from this, I think some discussion of near- and far-transfer effects is relevant here (e.g., Harris et al., 2020). This seems relevant to the authors’ proposal that the standard MOT task is not soccer-specific enough to find expertise effects.

d. A recent review of Neurotracker (3D MOT training) also seems relevant (Vater et al., 2021) which concludes that there is limited evidence for transfer of MOT training to actual sport skills. These findings seem to support the rationale for your study suggesting sport-specific MOT tasks should be considered.

2. The research question is interesting, and the primary objective is clearly stated (lines 48 – 49). However, I find it somewhat difficult to understand the authors’ hypothesis. My understanding from the introduction is that they think that soccer-expertise will only effect tracking performance on a soccer-specific task. Based on this, I would hypothesise an interaction effect: no difference in performance between the groups on the unstructured task but better performance by the soccer group on the structured task. I find the authors’ explanation of this somewhat confusing (lines 57 – 67) and think it could be written more clearly.

a. In particular, I find the use of ‘performance differential’ difficult to interpret (even though I think it is essentially describing the hypothesis I outline above). Since the authors go on to run an ANOVA, I think it makes more sense for the hypothesis to be explained in terms of an interaction.

3. I had several queries when reading the methods section of the experiments that are detailed below. The authors do recognise several of these points as limitations of their experiment in the discussion. Although this is good, I do think that they should elaborate on why, despite these limitations, they are still confident in the quality of the data and thus results presented here.

a. Were people asked whether they watched soccer/sport? It seems likely that these people would also display enhanced tracking abilities in sport scenarios despite not necessarily playing the sport themselves.

b. I wonder why the authors picked a defender’s viewpoint. What are the implications of this for players who are not defenders? I expect a striker-specific viewpoint to be very different from a defender-specific viewpoint.

c. There is quite some variability in the standard of the players. My assumption is that the movement trajectories were taken from professional games (lines 143, also please clarify whether this is the case) and I wonder what the implications of this are. Is it likely that district level soccer players are really familiar with the movement trajectories they were exposed to? If not, why would we expect them to show an advantage?

d. I would appreciate more detail on the data collection. This study was ran online so I wonder how the researchers ensured standardization of viewing factors (e.g., screen dimensions, refresh rates)? It is well-documented that things like tracking area size and speed of targets effect performance in MOT.

e. I wonder why the authors only used 15 trials in each condition. Based on my knowledge, this seems rather low compared to other MOT experiments.

f. I wonder why there was no ball. Presumably that seems like an important anchor in these football specific scenarios in that the ball-carrier dictates where people look and move?

4. I think it’s good that the authors made an example of task available online. However, this raises concerns about the task because, personally, I can’t see the players in the other half with sufficient acuity to track them. I do realise that I am missing an important component of the task whereby the participants could rotate their viewpoint. Nevertheless, I do struggle to visualize this (e.g., which axis do I rotate around/can I move to a ‘bird’s eye view’) so would appreciate an example of the possible viewpoint changes as well.

5. Following on from this, I wonder whether the authors could provide more justification for their decision to allow viewpoint changes. Research has shown that viewpoint changes have a large effect on tracking ability (e.g., Huff et al., 2009, Seiffert et al., 2005) so I think this addition to the task makes the interpretation of the data more complicated. I think the authors should consider the implications of allowing participants to change viewpoint throughout the manuscript. This additional component of the task arguably makes it less similar to soccer.

6. Since the viewpoint changes are an essential component of this study in my opinion, I wonder why the authors didn’t provide any data on this. Would the authors be happy to share this data and provide some insight into this?

7. The exploratory analysis using PCA and HCPC does not really fit with the experiment and I find interferes with the overall narrative of the paper. I wonder why the authors chose to include this analysis. Moreover, I find it difficult to understand how the features were calculated and the implications of this for interpretation. My understanding is that the features were expressed in terms of image coordinate (viewing angle) which would have been different across each participant and each scenario. Is it not possible that these results are therefore largely effected by viewing angles?

8. I find Figure 2 difficult to interpret and think more detail is needed. What does the colour coding on each panel represent? What are the axes of the middle panel? What are the units in each panel?

9. Throughout the manuscript, the authors often talk about ‘improvement’ which I find confusing. From my understanding, this is not a training study where participants were tested at different time-points and thus could show improvements, but rather a single testing session. I think the question being addressed is whether one of the groups is better.

10. The discussion says ‘All participants certainly did not comply with the instruction to select only those target-players whom they were confident enough to have successfully completed the tracking’ and that this ‘may have artificially improved the success rate of participants’ (lines 313 – 320). I am not entirely sure what the authors mean by this. Did they not check how many players were selected and verify whether these were correct or not?

Minor Comments

1. The third sentence of the abstract is rather long and difficult to follow.

2. I don’t follow the reasoning in lines 35 – 38. The authors suggest there is sport-specific gaze control so, presumably, there is also MOT-specific gaze control. The task/sport determines the gaze so I don’t find this result surprising.

3. I find ‘teammates’ more intuitive than partners in the context of soccer.

4. Line 284 – 285: ‘TEAM also tended to outperform NOTEAM but not significantly (p = .071)’. I think this an over interpretation of the results and should be removed.

6. PLOS authors have the option to publish the peer review history of their article (what does this mean?). If published, this will include your full peer review and any attached files.

Reviewer #1: No

Reviewer #2: No

---

## [Author Response · Author response to Decision Letter 0]

25 Mar 2022

The response to the editor and reviewers is available as an attachment to this submission ("Response to reviewers.pdf", and is copied here without formatting:

*****

Dear Editor, Dear Reviewers,

My colleagues and I thank you very much for your interest and insightful comments on our manuscript entitled “Visual tracking assessment in a soccer-specific virtual environment: a web-based study”. Please find attached the manuscript revised following your questions and remarks. We have carefully addressed each of the concerns or issues you have raised. All additions to the document are indicated in red in the manuscript with track changes. All deletions are also crossed out in red in the manuscript with track changes. All changes are also noted in this response to the reviewers and the lines mentioned refer to the manuscript file with the track changes. Additionally, we include all raw data underlying our results available in supplementary files (S3File: Visual tracking performance of each participant by experimental condition, S4File: Input data for exploratory analysis regarding the STRU condition and S5File: Input data for exploratory analysis regarding the UNSTRU condition). Thank you again for giving us the opportunity to defend our choices and for your comments that helped us improve the manuscript.

Kind regards,

Journal requirements

Please ensure that your manuscript meets PLOS ONE's style requirements, including those for file naming

Thank you for the comment. We made modifications to the figure files with the use of the PACE digital diagnostic tool to meet PLOS requirements.

Please amend your current ethics statement to address the following concerns:

Did participants provide their written or verbal informed consent to participate in this study?

Participants gave informed consent to participate in this study via an online form. After reading the information note and consent form, they had to tick a checkbox corresponding to the statement “I have read the information about the experiment and I agree to participate in the project, without any constraints or external pressure.” They were then asked to provide an email address and press a submit button to confirm their consent to participate. At the email address provided, they received an automated link that allowed them to log into the online application and participate in the study.

If consent was verbal, please explain i) why written consent was not obtained, ii) how you documented participant consent, and iii) whether the ethics committees/IRB approved this consent procedure.

i) Written consent was not obtained since the experiment was carried out on the internet. ii) Once on the website, they were invited to read information about the study presented in a PDF file embedded in a frame on the web page. They were able to download the PDF file if necessary. iii) The local ethic committees of the University Rennes 2 approved this consent procedure. 

For clarity in this regard, we have added in the revised manuscript:

line 234: “At the end of the form, participants agreed to participate by ticking the appropriate checkbox, entering their email address and clicking a submit button. They then received a link to access the online application to participate in the study.”

line 237: “The data collection procedure was validated by the local ethic committee of the University Rennes 2 (reference number: 2021-012).”

Thank you for stating in your Funding Statement: [This study was partially funded by the ANR within the framework of the PIA EUR DIGISPORT project (ANR-18-EURE-0022) and by the Region Bretagne.]

Please provide an amended statement that declares *all* the funding or sources of support (whether external or internal to your organization) received during this study, as detailed online in our guide for authors at http://journals.plos.org/plosone/s/submit-now. Please also include the statement “There was no additional external funding received for this study.” in your updated Funding Statement. Please include your amended Funding Statement within your cover letter. We will change the online submission form on your behalf.

Thank you for the comment. In this regard, we added in the submission process about the fundings:

“The funders had no role in study design, data collection and analysis, decision to publish, or preparation of the manuscript. There was no additional external funding received for this study.”

Please ensure that you include a title page within your main document. You should list all authors and all affiliations as per our author instructions and clearly indicate the corresponding author.

Thank you for the reminder, a title page has been added in the revised manuscript with a complete list of authors and affiliations. The corresponding author has also been listed.

Reviewer 1

The authors studied how an observer visually tracks moving elements in a sport environment. They reduce a bias present in MOT protocols by simulating the movements of players on a field thanks to player movements extracted from a database of real football matches.

The authors compare the tracking performances of football players, athletes who do not play football but another team-based sport, and non-sport-practising individuals. They also have participants track scenes where virtual players follow real game trajectories, and also when the virtual players follow non-natural movements. Using ANOVA and HCPC analyses, the authors wished to analyse what experimental factors influenced tracking in the virtual scenes.

The presentation of the study, its details and findings were well presented and easy to understand.

Thank you very much for all your comments and questions which we address in the following.

Is it possible that the fact that virtual players are showing no discerning features (beyond their team's colour) may have made the task harder than it would be in a real-case scenario? Does the choice of making all players identical reflect a belief of the authors that visual tracking in soccer is a process requiring very few visual features? I ask because the authors state their interest in approaching more natural conditions several times in the manuscript, but there was no discussion of realism related to visual features of the virtual players.

Thank you for your questions. We are not yet able to say whether the proposed task was more difficult than it would be in a real-world scenario. And the choice to make all players identical was made to match a multiple object tracking (MOT) task in which all objects are identical in appearance. Some evidence has supported the validity of this task to assess participants' ability to allocate visual attention simultaneously to multiple moving objects as a function of their sports practice (Faubert, 2013 ; Qiu & al., 2018 ; Harris & al., 2020 ; Jin & al., 2020). Our concern was only the naturalism of the position and movement of the virtual objects. North and colleagues (2016) compared the accuracy of high-skill and less-skill soccer players to anticipate a game situation outcome when viewing video-film and point light display formats. They observed that highly skilled soccer players made more accurate anticipations when watching a video-film or a point light display (without any discerning features). Thus, they suggested that the perception of structured collective behavior alone was sufficient to anticipate the outcome of the situation. We believed that only manipulating virtual players’ trajectory, and no other visual features was sufficient to address the outlined issue. In addition, we only had data on the trajectory of real-world players, but no data on their orientation or gestures. We preferred to simulate only simplified humanoid dummies without exact body composition because we felt that adding computer-generated body gestures would ultimately degrade the realism of the virtual players instead of improving it. Finally, one way to go further in what you meant would be to implement a Multiple-Identity-Task (MIT) task instead of a MOT task (Oksama & Hyöna, 2019). In MIT, every virtual player would have its own identity (e.g. color haircut, height and width, jersey number, …). To accurately answer your questions, it would certainly be interesting to test in the future whether the soccer players' advantage would be greater in MIT than in MOT.

In this regard, we have added these details in the revised manuscript:

line 62: North and al. 2016 reference have been added

lines 168-177: “North and colleagues (2016) compared the accuracy of high-skill and less-skill soccer players to anticipate a game situation outcome when viewing video-film and point light display formats [25]. They observed that high-skilled soccer players made more accurate anticipations when watching a video-film or a point light display (without any discerning features) [25]. Thus, they suggested that the perception of structured collective behaviors alone was sufficient to anticipate the outcome of the situation [25]. Only manipulating virtual players’ trajectory, and no other visual features, would be sufficient to improve the naturalism of the visual tracking task and to benefit the soccer players [28].”

Was it really necessary to give participants the possibility to pan the camera view? The authors point to this as a limit in the discussion section because some participants may have had issues manipulating the camera, on top of tracking elements in the scene. This is potential a bias that could have been easily avoided by restricting the movements of the virtual player to be within the camera viewport.

Thank you for this question and this comment. In fact, we have been clumsy in expressing this limit. This was confusing because allowing participants to rotate the point of view was a conscious choice that we explain in more detail in the revised manuscript (lines 270-291):

“On a real soccer field, players engage in active scanning behaviors, moving and rotating their bodies and heads, to gather information about their surroundings [43,44]. Jordet and colleagues suggested that this active visual exploration to overcome spatial field constraints has a positive effect on soccer-specific performance [45]. This component was addressed in the study by Ehmann and colleagues [46]. Visual tracking performance was assessed in 360° MOT scenarios that allowed participants to reorient themselves during the visual tracking task to mimic the spatial constraints of the soccer field [46]. They then observed age-related changes and the effect of soccer performance on the ability to track multiple moving virtual players in a 360° environment [47]. Thus, we deliberately chose to add this rotation component in an attempt to get closer to the constraints of the field, i.e., that the players to be monitored are moving all around and not just on a frontal plane.”

Please find an additional argument in the response to comment #5 of Reviewer #2. In addition, we asked participants about their frequency of using an Internet browser on a computer on a 5-level Likert scale ranging from 1 (Never) to 5 (Every day). 37, 11, 1 and 1 participants respectively answered 5, 4, 3 and 2. Thus, we confidently assume that participants were familiar with the use of a computer and its keyboard. In addition, during the 4-trial training session prior to the evaluation session, 2 trials required the vantage point to be rotated to successfully track the 4 target-players. These trials allowed participants to get used to the vantage point rotation. 

In this regard, we removed the paragraph at line 513 to moderate this limit in discussion of the revised manuscript. And we added this paragraph instead (lines 524-531):

“Other limits can be noted regarding the use of 2D stimuli that were constrained by the frame of a computer. First, depth information, which is important for tracking multiple objects in 3D space [53], was not available on the 2D screen. Second, the interaction medium that allowed exploration of the environment (i.e., viewpoint pivot with keyboard keys without translation) was poorly representative of the possibilities of visual exploration in the field [54]. This did not allow soccer players to engage more elaborated information gathering behaviors like body position adjustments [46].”

In our opinion, the actual limitation to point out was not the difficulty of using the keyboard, because in fact it was probably not that difficult, but rather the fact that the use of the keyboard was not representative enough of the visual exploration possibilities during a soccer game. Due to the lockdown during the COVID-19 pandemic, we had no choice but to conduct this study via the Internet and work with this limitation.

I understand that using a method such as HCPC allows for the study of many variables at once and to identify and characterise varied behaviours. But this appears rather post-hoc and nebulous to me in the interpretations made from the resulting data. It is complicated to measure and attribute the effects amongst the variables related to each group.

Thank you for this comment. What is important to note is that the situational features significantly associated with the clusters of hard-to-track target-players varied between the experimental conditions. This means that in order to understand how a situational feature affects tracking in the real world, it is important to use real-world visual tracking scenarios, rather than random scenarios. This would help to identify difficulty factors to manipulate in training programs. In this regard, we have shortened the description of the results and tried to focus our interpretation on the difference between the two experimental conditions on the returned clusters of hard-to-track target-players:

In the Results section, we removed lines 393-397, lines 400-404, lines 408-413, lines 432-437, lines 440-445 and lines 448-453.

And we respectively added instead:

line 397: “Cluster #1 consisted of 10 target-players that were associated with significantly lower Ratio.NoTeam, Ratio.Team, Ratio.Socc, Ind.DensityCol, Grp.Disp, Grp.Density and significantly higher Ind.DistTarget, Grp.Speed than the global average (S1_File)”

line 404: “Cluster #2 consisted of 16 target-players that were associated with significantly lower Grp.Disp, Ind.DistTarget, Ratio.Team, Ratio.Socc and with significantly higher Grp.Density, Grp.DensityCol, Ind.Density, Ind.DensityCol than the global average (S1_File). ”

line 413: “Cluster #3 consisted of 34 target-players that were associated with significantly lower Grp.DensityCol, Ind.Density, Grp.Density, Ind.DistTarget, Grp.Speed and with significantly higher Ratio.Team, Grp.Disp, Ratio.Socc, Ratio.NoTeam, Ind.Speed than the global average (S1_File).

line 437: “Cluster #1 consisted of 25 target-players that were associated with significantly lower Grp.Density, Grp.Disp, Grp.Speed, Grp.DensityCol, Ind.Density and significantly higher Ratio.Team, Ind.DistTarget, Ratio.Socc than the global average (S2_File).”

line 446: “Cluster #2 consisted of 24 target-players that were associated with significantly lower Grp.Disp, Ind.DistTarget and higher Grp.Density, Grp.Speed, Ratio.NoTeam, Ind.Density and Ratio.Socc than the global average (S2_File).”

line 453: “Cluster #3 consisted of 11 target-players that were associated with significantly lower Ratio.Socc, Ratio.Team, Ratio.NoTeam and significantly higher Grp.DensityCol, Grp.Speed than the global average (S2_File).”

In Discussion section: 

We added (lines 555-561): 

“This procedure allowed to describe the hard-to-track target-players in both experimental conditions (virtual players following soccer-specific or pseudo-random trajectories), while observing the associated situational features variations. Hard-to-track target-players were identified when the successful tracking ratio of a cluster was below the global average. Target-players in clusters #1 and #2 were considered as hard-to-track in STRU condition (Fig.4-B) and target-players in cluster #3 were considered as hard-to-track in UNSTRU condition (Fig.5-B).”

We removed the paragraph from line 562 to line 594 and we added this paragraph instead (lines 595- 615): 

“Regarding the STRU condition, having two distinct clusters of hard-to-track target-players suggests that tracking performance was not affected by a single situational feature but rather depended on specific combinations of them. On the one hand, hard-to-track target-players from cluster #1 were likely to be lost because they were far from the other three target-players (high Ind.DistTarget) which were grouped together (low Grp.Disp) and which moved rather fast (high Grp.Speed). These hard-to-track target-players illustrated the difficulty to gather information on dispersed field areas, especially when one area is a major source of interest. Participants probably did not divide their attention equally among the four target-players, but rather focused on the three grouped target-players that moved in a coordinated manner in the same area. The grouping strategy likely ensured the success of the tracking on at least three target-players [59,60]. Despite the ability to gather information on spread areas seemed to have a positive effect on soccer-specific performance on the field [45], the soccer players had the same difficulties as the other participants on these hard-to-track targets-players. On the other hand, soccer and other team sports players likely missed the target-players in cluster \\#2 due to visual interference induced by the density of the disruptors surrounding each of the target-players (high Grp.Density, Grp.DensityCol, Ind.Density and Ind.DensityCol). These results are consistent with the literature, which has observed a negative influence of density on visual tracking performance [8,37]. These hard-to-track targets-players reflected the difficulty in disentangling information in very dense situations where many players of both teams were located in the same field area.”

We made somes changes from line 616 to 622: “Regarding the UNSTRU condition, the hard-to-track target-players in cluster #3 were characterized by a very high density of same-colored disruptors around the other target-players (high Grp.DensityCol) involved in the same situation that moved fairly fast (high Grp.Speed).”

We removed line 625.

We have specified line 629: “(...) crowding and speed (...)”

We have also specified line 630: “These hard-to-track target-players were in fact missed not because of their own situational characteristics but because of those of other target-players”

We have also specified line 638: “(...) of same color disruptors (...)”

We removed line 639 and we added instead this paragraph (lines 640-672): 

“In summary, when comparing results across experimental conditions, the situational features associated with hard-to-track players differed. In STRU condition, isolated target-players (cluster #1) and target-players in very cluttered space (cluster #2) were more likely to be lost, whereas in UNSTRU condition, target-players had more chance to be lost when the others to monitor were moving fast and were crowded by same color disruptors. This result suggests that with real scenarios, the space distribution of virtual players had a stronger influence on visual tracking performance than with pseudo-random scenarios. Furthermore, as the initial and final locations of the virtual players were the same between a STRU situation and its associated UNSTRU situation, the only difference in the situational features of target-players arose from the distribution of the virtual players in space during the movement phase. This generated different clusters of target-players. On a soccer field, the players are not uniformly distributed on the field, but the density varies in different areas, depending on tactical objectives with respect to the location of the ball. This concern has already been investigated in the study by Vater and colleagues [37]. The authors observed that targets grouped in cluttered spaces attracted participants' gaze to reduce the negative effect of crowding [37]. Participants' gaze activity has not been recorded in this study, but they might also have allocated their foveal attention to grouped target-players in crowded spaces. In STRU condition, the influence of speed was not as clear. On the one hand, hard-to-track target-players in cluster #1 presented a significantly higher than global average Grp.Speed suggesting that speed feature may have had a negative effect on the visual tracking [38]. On the other hand, easy-to-track target-players (cluster #3) were target-players with high speed (Fig 4-B). Conversely in the UNSTRU condition, the speed features have a more obvious negative effect on visual tracking performance since the hard-to-track target-players were described with the highest Grp.Speed and Ind.Speed (Fig 5-B). Therefore, in contrast to the pseudo-random movements, the visual tracking performance with real-world movements have been more affected by features related to the spatial distribution of virtual players than by features related to speed. In addition, unlike in the STRU condition, no clusters clearly illustrated easy-to-track target-players in the UNSTRU condition. Apart from the 11 target-players in cluster #3, the remaining 49 had an equal probability of being successfully tracked. This suggests that real-world trajectories invited participants to focus more on certain target-players and leave others.”

Additionally, although fig. 5.A shows fairly separable clusters of target-players, that is not the case for fig. 4.A.

Thank you for your comment. It is indeed true that the clusters are less separable in Figure 4A . We add a sentence in the revised manuscript to warn the reader about this point at line 387: 

”For the sake of brevity, only the first two components are shown in Fig 4-A. The plot may be misleading due to the significant overlap of the three target player groups, but as a reminder, the third and fourth PCs were also retained for clustering.

I realise that the alternative would be to vary each variable independently and measure their resulting effect, therefore multiplicating the number of experiments needed to run. But considering the fact that the authors present HCPC as an exploratory tool, I would at least expect to read in the discussion section about what they would do in future work to study more precisely the specific behaviours the HCPC analysis highlighted.

Thank you for the comment. In this regard, we provide new ideas in the revised manuscript on what would be interesting to study in future work based on these results. We removed the paragraph from line 675 to line 688 and the paragraph from line 709 to 713. We added this paragraph instead (lines 688-706): 

“For future work on this issue, we recommend to not alter the movements of virtual players by individually manipulating each situational feature, but rather to use the situational features associated with hard-to-track target-players as selection factors for real-world game situations. In concrete terms, the game situation should be selected based on the density of players in a given area of the field and target-players should be selected based on the clutter in their neighborhood. This would allow the study of the influence of the spatial dispersion of players on the visual tracking task when facing real world movements. In addition, this study reported an average speed of real-world target-players of 3.3 ± 1.6 °/s, which was lower than the range of speed studied in other sports performance studies [1,18,19]. Tracking players who run at a high pace might be more difficult than tracking players who walk slowly [38], but we encourage future research on the relationship between sport performance and visual tracking performance to also rely on spatial features and not just the speed feature. Although visual tracking training improved the ability to track multiple fast objects [15,31,33,35] transfer to a sport-specific task may not have occurred because of the lack of correspondence of field stimuli [28]. Instead of training the ability to track objects in a small space (2D screen) that move faster than in the real world, it would be interesting to train the ability to track objects in a large [46] and cluttered [37] space, to see if higher transfer benefits to a sport-specific task would occur. ” 

p.2 "players movements" -> "player movements"

Thank you for the comment and the error has been corrected.

p.3 "allow participants to access to the" -> "allow participants to access the"

Thank you for the comment and the error has been corrected.

p.6 "not difference" -> "no difference"

Thank you for the comment and the error has been corrected.

Reviewer 2

This paper investigates whether playing a team sport effects tracking performance in a visual tracking task using movement trajectories taken from soccer matches (i.e., structured). The authors compared the tracking performance of soccer players, other team sport players, and non-team sports players. They also compared performance on structured and unstructured movement trajectories. The authors did not find that soccer players displayed better tracking performance when facing the structured movement trajectories taken from real soccer games compared with the other two groups. Since the results did not fit with the authors’ hypothesis, they suggest an immersive design that captures the soccer field more closely should be used in future research.

The manuscript fits nicely within the scope of PLOS ONE and addresses an interesting question. However, there are several methodological limitations (many of which the authors outline themselves in the discussion) that make me wonder about the quality of the data and makes interpretation of the results somewhat difficult.

Thank you very much for all your comments and questions which we address in the following.

The introduction is clear and includes some relevant literature. However, I do think it could be more focussed towards the specific questions the authors are interested in addressing.

Expertise effects could be discussed in more detail. Quite some relevant research has been done into video game players (e.g., Dye & Bavelier, 2010; Green & Bavelier, 2006; Sekuler et al., 2008).

Thank you for the remark and the proposed litterature. In this regard, the suggested references have been added in the revised manuscript (lines 19-38) : 

“Like radar operators, action video game players must deal with multiple objects in complex and dynamic virtual environments. Expertise effect on MOT was also observed in the video games field, where action video game players demonstrated higher tracking performance than non-video game players [10-13], and where professional video games players also demonstrated higher tracking performance than amateur video games players [14]. In addition, Green and Bavelier observed that tracking performance was improved in non-video game players when they received regular training on a first-person view action video game [11]. The authors suggested that this improvement in tracking performance was enabled by video game-induced improvements in some aspects of visual short-term memory [11]. In addition, playing action video games has been found to have an effect on both adult [12] and child populations [13]. We can note that age-related changes were observed in the study by Dye and Bavelier in which tracking performance increased in young people as a function of age (7-10, 11-13, 14-17, and 18-22 years) and in the study by Sekuler and colleagues in which the tracking performance of young adults (approximately 20.6 years) was higher than that of older adults (approximately 75.3 years) [12, 13]. Like action video games, some sports require to attend to multiple environmental sources of information [10]. Trick and colleagues compared the tracking performance of children in and out of action sports (field hockey, soccer, ultimate, martial arts), and observed a marginal group effect in favor of action sports group [10].”

I would appreciate some discussion on why these conflicting results regarding expertise differences have been found. For example, why does expertise in some sports seem to lead to better MOT performance (10 – 13) but not in others (15)?

Thank you for this comment. In this regard, clarifications have been made (lines 45-54):

“Qiu and colleagues observed an expertise effect when at least three targets were to be tracked, but they observed no inter-participants differences when only two targets were tracked [17]. Another explanation would be that visual tracking ability does not seem to develop linearly with the level of practice [17]. In the initial visual tracking assessment of Faubert's study [15], and in the study of Qiu and colleagues [17], differences were observed between elite athletes and intermediate athletes, but not between intermediate athletes and non-athletes. Regarding the standard of handball players in the study by Memmert and colleagues [20],it is possible that they were more of an intermediate standard than an elite standard even if they practiced for more than 10 years [21]. Finally, (...)”

Following on from this, I think some discussion of near- and far-transfer effects is relevant here (e.g., Harris et al., 2020). This seems relevant to the authors’ proposal that the standard MOT task is not soccer-specific enough to find expertise effects.

A recent review of Neurotracker (3D MOT training) also seems relevant (Vater et al., 2021) which concludes that there is limited evidence for transfer of MOT training to actual sport skills. These findings seem to support the rationale for your study suggesting sport-specific MOT tasks should be considered.

Thank you very much for highlighting these two points. We are responding to both comments at the same time because we believe they are related. The suggested references have been added in the revised manuscript (lines 84-94):

“This representativeness issue was also raised with respect to the question of transferring the benefits of generic perceptual-cognitive training to actual performance in the field [28-30]. For training based on a generic visual tracking task, Romeas and colleagues observed promising initial results, where soccer players' passing decision making on small-sided-games increased after training [30]. However, limitations regarding the small sample size suggested that the results needed to be replicated to confirm this conclusion [31]. Then, with a larger sample size, Harenger and colleagues also studied the transfer of benefits from visual tracking training, but they did not observe significant improvement in a soccer-specific decision making task [32]. Other contrasting results were revealed, indicating the lack of transfer of generic visual tracking training to tasks representative of actual military [33] or sporting activities [34]. Finally, the recent systematic review by Vater, Gray and Holcombe alerted to the lack of evidence supporting a real transfer of generic-visual tracking based training to sport-specific performance [35].”

 The research question is interesting, and the primary objective is clearly stated (lines 48 – 49). However, I find it somewhat difficult to understand the authors’ hypothesis. My understanding from the introduction is that they think that soccer-expertise will only effect tracking performance on a soccer-specific task. Based on this, I would hypothesise an interaction effect: no difference in performance between the groups on the unstructured task but better performance by the soccer group on the structured task. I find the authors’ explanation of this somewhat confusing (lines 57 – 67) and think it could be written more clearly.

In particular, I find the use of ‘performance differential’ difficult to interpret (even though I think it is essentially describing the hypothesis I outline above). Since the authors go on to run an ANOVA, I think it makes more sense for the hypothesis to be explained in terms of an interaction.

Thank you very much for pointing out this confusion. In this regard, we have clarified our hypothesis in the revised manuscript:

The paragraph from line 147 to 158 has been removed.

This paragraph have been added instead (lines 158-168):

“As team sport players demonstrated higher visual tracking performance than non-team sport players in generic MOT tasks [1, 17-19], we expected to observe a main group effect such that SOCC and TEAM would outperform NoTEAM regardless of condition. Then, and of most interest, we expected to observe an interaction effect between group and condition, such that the difference in visual tracking performance between SOCC and the other two groups was greater in the STRU condition than in the UNSTRU condition. Because of their knowledge and experience, soccer players could have an advantage over other participants when the tracking task involved soccer-specific movements. TEAM group was added in the investigation to see if the soccer-specific movements used are beneficial only to soccer players and not to other team sports players.”

I had several queries when reading the methods section of the experiments that are detailed below. The authors do recognise several of these points as limitations of their experiment in the discussion. Although this is good, I do think that they should elaborate on why, despite these limitations, they are still confident in the quality of the data and thus results presented here.

Were people asked whether they watched soccer/sport? It seems likely that these people would also display enhanced tracking abilities in sport scenarios despite not necessarily playing the sport themselves.

Thank you for this interesting question. This is exactly what we asked ourselves before the experiment. In the online application, participants completed a questionnaire before conducting the evaluation. Participants were asked to indicate on a 5-point likert scale how often they watch soccer games (Never - Once a year - Once a month - Once a week - Several times a week). 

To test whether soccer games viewing frequency affects visual tracking performance, we grouped participants according to their response and we compared the mean tracking performance in both conditions. 

Viewing Frequency

Mean Tracking performance ± SD STRU 

Mean Tracking performance ± SD UNSTRU 

Never (n=6)

0.667 ± 0.097

0.622 ± 0.130

Once a Year (n=19) 

0.689 ± 0.125

0.647 ± 0.104

Once a Month (n=7) 

0.702 ± 0.130

0.624 ± 0.117

Once a Week (n=12) 

0.737 ± 0.122

0.664 ± 0.119

Several times a Week (n=6) 

0.756 ± 0.109

0.706 ± 0.113

In both conditions, mean visual tracking performance appeared to increase with the frequency of soccer game viewing, but we assumed that this effect was not significant due to the large standard deviation. 

I wonder why the authors picked a defender’s viewpoint. What are the implications of this for players who are not defenders? I expect a striker-specific viewpoint to be very different from a defender-specific viewpoint.

Thank you for this question. Studies by Mangine and colleagues (2014) and Martin and colleagues (2017) observed differences in visual tracking performance between frontcourt and backcourt basketball players and between forward and back rugby players respectively. By the way, we omitted the reference to Mangine and colleagues in the original manuscript, so we have added it in the revised version at line 44. Attentional demands in the field differ depending on the role played. We chose the defender's point of view because we believe that players in the defender position are more likely to monitor multiple teammates and opponents in complex and dynamic scenes. A similar defender point of view was used in studies by Roca and colleagues (2011, 2013) to investigate participants' anticipation when viewing videos of soccer game situations. Looking at the literature (Mangine et al., 2014; Martin et al., 2017), it can be assumed that defenders would perform better than strikers in the proposed task. It would be interesting to see if playing position would have also influenced tracking performance in this study, but the sample size of soccer players did not allow for this analysis (3 strikers, 6 midfielders, 6 defenders, and 1 participant did not fill in his position).

Here are the mean visual tracking performance (and standard deviation) for the 3 groups of position.

Position

 Mean visual tracking performance ± SD

Striker (n=3)

0.717 ± 0.022

Defender (n=6)

0.719 ± 0.130

Midfielders (n=6)

0.742 ± 0.115

There is quite some variability in the standard of the players. My assumption is that the movement trajectories were taken from professional games (lines 143, also please clarify whether this is the case) and I wonder what the implications of this are. Is it likely that district level soccer players are really familiar with the movement trajectories they were exposed to? If not, why would we expect them to show an advantage?

Thank you for this comment. We do not know from which actual soccer games the data were provided. These anonymized data are made available to the research community and come from high-level games, the only fields equipped with these tracking systems. We clarify this point in the revised manuscript (line 295): 

“It was unclear from which actual soccer games the data were provided, but the data came from high-level championships, as this level of competition was the most likely to have a tracking system.”

Thank you for the questions that followed. Although game strategies may differ by level of competition, we strongly assume that there are common traits in the logic of the movements and thus that even district players were familiar with these movements. These common traits could be associated with the rules of the game (allowed playing space, offside rule) and the organization of teams into playing positions. In this regard, we add this assumption in the revised manuscript (line 298)): 

“In addition, we assumed that situations have still been familiar to district level soccer players, even if the situations come from high-level championships, because there is a common logic of play between the different levels of competition.”

I would appreciate more detail on the data collection. This study was ran online so I wonder how the researchers ensured standardization of viewing factors (e.g., screen dimensions, refresh rates)? It is well-documented that things like tracking area size and speed of targets effect performance in MOT.

Thank you for the question. To ensure standardization of viewing, we asked participants to position their eyes at a distance from their screen equivalent to its width. We also asked them to use a computer to perform the experiment, not a smartphone or tablet. This last point has been added in the revised manuscript at line 245:

“(...) to use a computer (no smartphone or tablet) and (...)”

We did not collect any information according to the refresh rate. In our pre-experimentation, we tested the online application with different computers and browsers. We did not experience any latency during the visual tracking task. We also did not receive any complaints from participants about latency. Of course, these arguments are not implacable, but we assume that little, if any, latency occurred during the experiment. Regarding screen dimensions, the task could only be performed in full screen mode and we collected the width and height of the participants' browser window. The height of the windows varied from 578 to 1137 pixels and the width of windows varied from 960 to 1920 pixels. Pearson’s correlation between mean tracking performance and screen height was 0.141 (p=0.425). Pearson’s correlation between mean tracking performance and screen width was -0.360 (p=0.306). Therefore, we assumed that the tracking performance was not so much influenced by the size of the participants' computer screen.

We take advantage of this section on data collection to inform the reviewers that finally we decide not to indicate the URL of the online application in the manuscript since it will not be available soon. This part of sentence has therefore been removed (line 230): “(website url was removed for anonymous reviewing).”

I wonder why the authors only used 15 trials in each condition. Based on my knowledge, this seems rather low compared to other MOT experiments.

Thank you for this comment. We agree that this was rather low compared to many other MOT experiments (e.g. 20 trials in Martin and al., 2017 ; 24 trials per condition in Qiu and al., 2018 ; 30 trials per condition in Jin & al., 2020). We chose this low number of trials to encourage participants to stay focused during the experiment. In the study by Harris and colleagues (2020), which compared 3 speed conditions and 3 target number conditions, participants completed 3 test blocks of 9 trials. The authors observed a significant difference in visual tracking performance between participants who played sports with high tracking demands and participants who did not (Harris & al., 2020). Therefore, we assumed that 15 trials in each condition would be sufficient in this investigation.

I wonder why there was no ball. Presumably that seems like an important anchor in these football specific scenarios in that the ball-carrier dictates where people look and move?

Thank you for this question. We also believe that the ball and the ball-carrier are important factors in attracting the visual attention of soccer players. We removed the ball from the situations because we felt that its influence would be different in the soccer-specific scenarios than in the pseudo-random scenarios. In the soccer-specific scenarios, the presence of the ball would probably have helped to perceive structured collective behaviors. North and colleagues observed that soccer players were more accurate in recognizing familiar soccer situations presented in point light displays when the ball was present than when it was not (2017). But we were not able to estimate how the behavior of the ball in pseudo-random scenarios would or would not have helped participants. Therefore, we decided to remove the ball from both experimental conditions. However, to further your point, we are currently conducting an experiment in which we are evaluating the tracking performance of soccer players when faced with soccer-specific situations with and without a ball. If you wish, we will gladly share the main results with you.

I think it’s good that the authors made an example of task available online. However, this raises concerns about the task because, personally, I can’t see the players in the other half with sufficient acuity to track them. I do realise that I am missing an important component of the task whereby the participants could rotate their viewpoint. Nevertheless, I do struggle to visualize this (e.g., which axis do I rotate around/can I move to a ‘bird’s eye view’) so would appreciate an example of the possible viewpoint changes as well.

Thank you for the comment. Regarding the concern about the resolution, the quality of the video was deteriorated due to a file size issue. We also recall that the task was performed in full screen mode, which allowed for better viewing than the video. The supporting video file has been updated according to your point. In this regard, clarification has also been done in the revised manuscript line (263): 

“In order to keep track of them, the participant could only rotate the vantage point around the vertical axis, i.e. to the left or right, using the corresponding directional arrow keys. No other modification of the vantage point was possible.”

Following on from this, I wonder whether the authors could provide more justification for their decision to allow viewpoint changes. Research has shown that viewpoint changes have a large effect on tracking ability (e.g., Huff et al., 2009, Seiffert et al., 2005) so I think this addition to the task makes the interpretation of the data more complicated. I think the authors should consider the implications of allowing participants to change viewpoint throughout the manuscript. This additional component of the task arguably makes it less similar to soccer.

Thank you very much for this comment. We agree that the original manuscript lacked arguments on this point. We also thank you for the suggested references. We have added one of them in the revised manuscript to take into account the influence of the change of viewpoint on visual tracking performance (Huff et al., 2009, Thomas and Seiffert, 2010) and we added this paragraph in the revised manuscript to justify why we thought this component of the task was important (lines 270-291):

“Previous studies have shown that changes in vantage point affect participants' visual tracking performance [41, 42]. In the study by Huff and colleagues, participants' vantage point abruptly rotated around the scene during the visual tracking task [41]. In Thomas and Seiffert's study, participants rotated around the scene passively or actively [42]. In both studies, tracking performance was reduced when participants changed their vantage point [41, 42]. Because self-motion interferes with multiple object tracking [42], allowing the pivot of the vantage point probably affected participant tracking performance in this study as well. However, the difference here was that the participants did not rotate around the scene, but reorient themselves to track the players as they would on a real soccer field. They controlled the rotations themselves and did so to successfully complete the visual tracking task. On a real soccer field, players engage in active scanning behaviors, moving and rotating their bodies and heads, to gather information about their surroundings [43,44]. Jordet and colleagues suggested that this active visual exploration to overcome spatial field constraints has a positive effect on soccer-specific performance [45]. This component was addressed in the study by Ehmann and colleagues [46]. Visual tracking performance was assessed in 360° MOT scenarios that allowed participants to reorient themselves during the visual tracking task to mimic the spatial constraints of the soccer field [46]. They then observed age-related changes and the effect of soccer performance on the ability to track multiple moving virtual players in a 360° environment [47]. Thus, we deliberately chose to add this rotation component in an attempt to get closer to the constraints of the field, i.e., that the players to be monitored are moving all around and not just on a frontal plane.”

The response to comment #2 from reviewer #1 can possibly be used as a supplemental response.

In Discussion, we also added this paragraph (lines 531- 540): 

“Abrupt changes in vantage point and rotation around the scene during a visual tracking task was observed to have a negative effect on visual tracking performance [41,42]. The rotation of the vantage point generated a very short period during which it was not possible to collect information, as during an ocular saccade [37]. It would also likely have affected the visual tracking performance of the participants in this study but, like the eye saccade, it was a behavior controlled by the participant to adequately explore the virtual environment. We strongly believe that it is necessary to retain this component of vantage point adjustment in the investigation of tracking performance of soccer players, as it is an inherent behavior of soccer practice. However, it seems preferable to use more natural means of exploration [54].”

Since the viewpoint changes are an essential component of this study in my opinion, I wonder why the authors didn’t provide any data on this. Would the authors be happy to share this data and provide some insight into this?

Thank you for the comment. Unfortunately, we did not record the rotations from the participants' vantage point during their evaluation. How soccer players visually explore their environment to successfully track multiple teammates and opponents is an interesting question. We are running another investigation on this question and we hope to share some leads on this topic soon.

The exploratory analysis using PCA and HCPC does not really fit with the experiment and I find interferes with the overall narrative of the paper. I wonder why the authors chose to include this analysis. Moreover, I find it difficult to understand how the features were calculated and the implications of this for interpretation. My understanding is that the features were expressed in terms of image coordinate (viewing angle) which would have been different across each participant and each scenario. Is it not possible that these results are therefore largely effected by viewing angles?

Thank you very much for your relevant comment. 

First, after rereading the original manuscript, we agree on the lack of arguments concerning the interest of this analysis. We also made substantial changes to the abstract and the manuscript to more seamlessly integrate the narrative of the exploratory analysis.

We removed the paragraph from line 193 to line 206.

We added:

Lines 62-73: “This lack of representativeness of the visual tracking task in relation to sport-specific scenarios is concerning in two ways. First, one might think that visual tracking performance measured in the laboratory with non-sport specific movements only partially reflects the actual ability of players to track multiple teammates and opponents moving across the field according to the logic of the game. Mann and colleagues' systematic review reported that the effect of perceptual-cognitive expertise was modulated by task representativeness [26]. Differences in visual tracking performance between participants based on their level of competition might be greater if the task tapped on more specific aspects of the studied domain of expertise [20].”

Lines 95: “This problem may stem from the lack of correspondence between the stimuli perceived in training and those in the real world [28] Subsequently, a second concern may be raised regarding the features manipulated to stimulate the visual tracking ability.”

Lines 98 - 129: “The second concerns lies in the relationship between tracking performance and sport performance that has been studied by manipulating situational features such as the number of targets [17, 29], object velocity [1,15,16,18,19] or tracking duration [19]. Manipulating these situational features one at a time allowed to test the participants' fundamental perceptual-cognitive abilities in a visual tracking task [3, 4] and was only possible with non real-world movements scenarios. However, it is questionable whether the situational feature manipulation was consistent with the situational features that can complicate the tracking task in real-world scenarios. On the one one hand, the current visual tracking training is aimed at increasing the speed threshold at which participants are able to successfully track multiple objects [15, 31, 33, 35]. Near transfer have been observed to other MOT task and mid-level transfer have been observed to working memory task [34] but, as mentioned earlier, the transfer of benefits to more sport-specific tasks was not so obvious [30]. In relation to the issue of transfer, it is questionable if, from a functional point of view, training the ability to follow faster and faster objects really allows one to adequately distribute one's attention in space in order to follow several players scattered on the field. On the other hand, regarding the study of the relationship between sports performance and visual tracking performance, it is questionable whether the objects' velocity used to observe differences between participants [1, 15, 16, 18, 19] was consistent with the velocity of objects to monitor in real-world. Depending on the study, the apparent velocity of moving objects was 16.8, 21.2 and 25.5 deg/s [19], or was7.4, 9.9 and 12.4 deg/s [18], or was 5, 10 and 15 deg/s [1]. Jin and colleagues did not observe tracking performance differences between basketball players and non-basketball players when objects moved at 5 deg/s [1], although this velocity seems reasonable with the constraints of the field since some of players to monitor can move at low speed or even be immobile. If the situational features used in the laboratory are not consistent with those of the real world, one may wonder about the generalization of the results to the field and their practical implications [28]. Speed is not the only situational feature that makes it difficult to track multiple objects in real-world scenarios. Beyond the speed factor itself, increasing the speed of the objects increases the spatial interference between the targets and the disruptors, which increases the difficulty of tracking [36]. Crowding can also be a situational feature that can complicate tracking in real-world scenarios [37].”

Line 130: “To address these two concerns, more sport-specific visual tracking tasks could be considered.”

Line 135: “Therefore, in this paper, a multiple players tracking task was proposed to investigate the relationship between sports practice and visual tracking performance in a virtual soccer environment. Two objectives arose from the above concerns. The primary objective was to study the interaction effect of sport practice and the specificity of virtual players movements on tracking performance.”

Lines 176-192: “The secondary objective of this study was to identify the situational features that have influenced the participants' performance in the proposed virtual players tracking task. From this perspective, the manipulation of situational characteristics was not possible with real scenarios without distorting them. If we substantially increased the velocity of the players to see how the velocity feature affected tracking performance, the real-world scenario would not have been so real. To preserve the naturalness of real-world scenarios, we conducted an original exploratory analysis with a clustering of virtual players to track, i.e. target-players, based on their successful tracking ratios and situational features identified as affecting the task. Because the target-players had their own situational features, we assumed that the probability of being correctly tracked would not be equal among them. Some targets should therefore be more difficult to track than others. The exploratory analysis aimed to describe the situational features associated with hard-to-track target-players. This analysis was applied to the STRU and UNSTRU conditions to see whether the hard-to-track target-players were described with a different set of situational features when virtual players were driven with real-world or pseudo-random trajectories.”

In the abstract: 

we removed (line 3): “ (...) stimuli, whereas the environmental conditions should be considered to investigate this performance factor.”

we added instead (lines 3-8): “(...) scenarios leading in two major concerns. First, the methods used probably only partially reflects the actual ability to track players on the field. Second, it is unclear whether the situational features manipulated to stimulate visual tracking ability match those that make it difficult to track real players.”

we removed lines 30-38 and we added instead (lines 38-46): “Regarding the second concern, an original exploratory analysis based on Hierarchical Clustering on Principal Components was conducted to compare the situational features associated with hard-to-track virtual players in soccer-specific or pseudo-random scenarios. it revealed differences in the situational feature sets associated with hard-to-track players based on scenario type. Essentially with soccer-specific scenarios, how the virtual players were distributed in space appeared to have a significant influence on visual tracking performance. These results highlight the need to consider real-world scenarios to understand what makes tracking multiple players difficult.”

Second, we agree that the use of the term "image coordinates" in the original manuscript was misleading. As mentioned in the modified paragraph, the viewing angle of a virtual player was computed from its coordinates and those of the participants' vantage point on the virtual field. In this way, the situational features were the same for all participants, regardless of how they controlled the vantage point. We clarified how the situational features were calculated in the revised manuscript (lines 341- 348):

“The coordinates of the vantage point of the participants and the coordinates of each virtual player with respect to the field were used to compute for each virtual player, an angle between the longitudinal axis of the field and the line connecting the virtual player to the vantage point. These angles were used to compute the situational features listed in Table 1. As participants had exactly the same vantage point in the virtual world, the perceived situational features were the same for all participants.”

In the discussion, we also removed the paragraph from line 708 to line 712, which had certainly led to confusion. We added instead (lines 712-717):

“In addition, features were expressed in terms of viewing angle and not in terms of field plane coordinates, which did not account for the absolute distance between a virtual player and the participant’s vantage point. The perceived size of a target-player decreased with its distance to the observer's location, potentially increasing the difficulty of the tracking task [61]. In future investigations, it would make sense to add scene coordinate features in order to better appreciate the situational features of players moving on the soccer field.”

I find Figure 2 difficult to interpret and think more detail is needed. What does the colour coding on each panel represent? What are the axes of the middle panel? What are the units in each panel?

Thank you for the comment and the questions. The description of the figure has been detailed to clarify this point:

“In each plot, the axes represent the coordinates of the soccer field in meters with the origin at the center of the field. Ticks and units have been removed from the x and y axis of the central panel to avoid redundancy with the other two panels.”

“The orange trajectories correspond to the players of the team carrying the ball. The blue trajectories are those of the players of the opposing team.”

“The initial and final positions are gray dots. Each final position is connected to its initial position by a gray line, just for this illustration.”

Throughout the manuscript, the authors often talk about ‘improvement’ which I find confusing. From my understanding, this is not a training study where participants were tested at different time-points and thus could show improvements, but rather a single testing session. I think the question being addressed is whether one of the groups is better.

Thank you for the comment. In this regard, we made somes changes:

The line 491 in discussion has been removed and this sentence have been added instead (line 493):

“Contrary to our expectation, visual tracking performance difference between soccer players and other participants was not greater when faced with stimuli involving soccer-specific movements compared to stimuli involving pseudo-random movements.”

The line 731 has been removed from the conclusion and this sentence has been added instead (733):

“The overall analysis did not support the expectation that the difference in tracking performance between soccer players and other participants would be greater when faced with stimuli directed by soccer-specific trajectories.”

The sentence at line 9 has been removed from the abstract and this sentence has been added instead (line 13 - 19):

“In this study, a visual tracking task was proposed, in which participants had to track multiple virtual players on a virtual soccer field. The virtual players moved according to either real or pseudo-random trajectories. (...). Regarding the first concern, the visual tracking performance of players in soccer, other team sports, and non-team sports was compared to see if differences between groups varied with the use of soccer-specific or pseudo-random movements.”

The sentence line 19 has been removed from the abstract and this sentence has been added instead:

“Contrary to our assumption, the ANOVA did not reveal a greater tracking performance difference between soccer players and the two other groups when facing stimuli featuring movements from actual soccer games compared to stimuli featuring pseudo-random ones.”

The sentence at line 23 has been removed from the abstract and this one has been added instead (line 25): 

“Directing virtual players with real-world trajectories did not appear to be sufficient to allow soccer players to use soccer-specific knowledge in their visual tracking.”

The discussion says ‘All participants certainly did not comply with the instruction to select only those target-players whom they were confident enough to have successfully completed the tracking’ and that this ‘may have artificially improved the success rate of participants’ (lines 313 – 320). I am not entirely sure what the authors mean by this. Did they not check how many players were selected and verify whether these were correct or not?

Thank you for the comment and the question. We acknowledge that the sentence made this limitation rather blunt. What we meant was that, to reduce the effect of chance in selecting target-players at the end of a trial, we asked participants to select only those target-players for whom they were certain to have completed the tracking. As a result, some participants did select only 3 or 2 target-players if they lost 1 or 2 during the trial. However, other participants ignored this instruction and selected 4 virtual players each time. This means that these participants were slightly more likely to successfully select a target player. We recognize that this instruction ultimately constituted a bias in the study rather than being helpful. But we assume that this was a minor bias since, in addition to the 4 target-players, there were still 17 disruptive virtual players with whom it was possible to be wrong. Thus, for participants who selected a correct target player after hesitating between two or three nearby virtual players, we can consider that their tracking ability still gave them a high probability of giving a correct answer. Guessing certainly had a very small impact on participants' visual tracking performance.

In this regard, we removed the line 505 in the revised manuscript and this sentence has been added instead (line 507 ):

“However, we assume that this bias induced by the possibility of guessing the target-players during selection was minor since the probability of guessing the correct targets in the middle of 17 distractors is low by pure chance. Moreover, finding a real target-player after guessing among some nearby virtual players still reflected a good level of visual tracking ability.”

The third sentence of the abstract is rather long and difficult to follow.

Thank you for the comment. In this regard, this sentence has been broken down and the change has been indicated in response to comment 9.

The first sentence has been modified to respect the number of words allowed in the abstract.

I don’t follow the reasoning in lines 35 – 38. The authors suggest there is sport-specific gaze control so, presumably, there is also MOT-specific gaze control. The task/sport determines the gaze so I don’t find this result surprising.

Thank you for the comment. In this regard, the paragraph has been updated (line 75):

“Perceptual-cognitive expertise is supported by effective gaze control [26, 27], which is why Harris and colleagues hypothesized that a different gaze strategy would have explained the superior tracking performance of participants involved in high tracking sports activity compared to participants who were not [18]. But contrary to their expectations, no differences in gaze strategy between participants were observed [18].”

I find ‘teammates’ more intuitive than partners in the context of soccer.

Thank you for the comment. In this regard, we have made the necessary changes.

Line 284 – 285: ‘TEAM also tended to outperform NOTEAM but not significantly (p = .071)’. I think this an over interpretation of the results and should be removed.

Thank you for the comment. In this regard, this sentence has been removed from the revised manuscript.

---

## [Decision Letter · Decision Letter 1]

18 Apr 2022

PONE-D-21-39830R1Visual tracking assessment in a soccer-specific virtual environment: a web-based studyPLOS ONE

Dear Dr. VU,

Thank you for submitting your manuscript to PLOS ONE. After careful consideration, we feel that it has merit but does not fully meet PLOS ONE’s publication criteria as it currently stands. Therefore, we invite you to submit a revised version of the manuscript that addresses the points raised during the review process.

Of the two reviewers who assessed your original submission, only Reviewer 1 was available to assess your revised manuscript. I have thus assessed your responses to the comments raised by Reviewer 2 myself. Reviewer 1 is satisfied with your responses to his comments. For the most part, I also think you have appropriately addressed the comments from Reviewer 2 in your rebuttal letter. However, some of your responses have not been incorporated in the revised manuscript, and some control analyses are still missing. In my comments below I thus ask you to perform these analyses and incorporate all your responses into your manuscript. Finally, I appreciate that you have added data files S3, S4 and S5 to the submission. I kindly ask you to also specify and describe, in the “Supporting information” section of the manuscript, what data are contained in supplementary files S3, S4, and S5.

We look forward to receiving your revised manuscript.

Kind regards,

Guido Maiello

Academic Editor

PLOS ONE

Additional Editor Comments (if provided):

Reviewer 2 Point 3a:

In your response to this reviewer comment, you present a table showing tracking performance as function of soccer viewing frequency. You state that even though “mean visual tracking performance appeared to increase with the frequency of soccer game viewing, but we assumed that this effect was not significant due to the large standard deviation”. I have plotted the data you presented, and computed 95% confidence intervals from your reported standard deviations and sample sizes. From these, it looks like there could be a significant effect, as in a few instances the means and confidence intervals of tracking performance across soccer viewing frequency do not overlap. Whichever the case, I suggest it is always best to verify ones’ assumptions when possible. I thus ask you to explicitly test whether a statistical statistically significant effect exists in these data. Further, I believe the reviewer’s main question was whether the observed differences between study groups could be explained by soccer viewing frequency. Thus it would be useful to test and report whether the three study groups differed significantly in soccer viewing frequency (e.g. by running a one-way ANOVA on viewing frequency, with sport practice as the between-subjects main effect. If the groups differ in viewing frequency the same way they differ in tracking performance, then this potential confound should be included and discussed in the discussion section of the manuscript.

Moreover, when a reviewer brings up a question, it is good practice to address it both in the response letter as well as in the main manuscript, since future readers may have the same question. I thus ask you to incorporate your response to this reviewer comment in the main manuscript. This could be simply a few lines in the methods or the discussion section of the manuscript, where you point out the issue and report the result of your control analyses (once you have performed them).

Reviewer 2 point 3b: As above, I ask you to report the result of these control analyses in the main manuscript.

Reviewer 2 point 3d: I suggest you should include in the manuscript these considerations and control analyses regarding frame rate and screen resolution. Perhaps you should also specify that by asking participants to position their eyes at a distance from the screen equivalent to its width, the screen should have subtended approximately 53 degrees of visual angle for all participants. For future reference, you should be aware that there exist validated methods to control viewing distance/stimulus size for online experiments, e.g.:

Li, Q., Joo, S. J., Yeatman, J. D., & Reinecke, K. (2020). Controlling for participants’ viewing distance in large-scale, psychophysical online experiments using a virtual chinrest. Scientific Reports, 10(1), 1-11

Lago, M. A. (2021). SimplePhy: An open-source tool for quick online perception experiments. Behavior Research Methods, 53(4), 1669-1676

Reviewer 2 point 3e: Please add a few short sentences in the methods section of the manuscript explaining your reasoning in selecting the number of trials.

Reviewer 2 point 3f: please include these considerations in the discussion section of the manuscript.

Line 531: “Abrupt changes … were observed”

Reviewers' comments:

Reviewer's Responses to Questions

**Comments to the Author**

1. If the authors have adequately addressed your comments raised in a previous round of review and you feel that this manuscript is now acceptable for publication, you may indicate that here to bypass the “Comments to the Author” section, enter your conflict of interest statement in the “Confidential to Editor” section, and submit your "Accept" recommendation.

Reviewer #1: All comments have been addressed

2. Is the manuscript technically sound, and do the data support the conclusions?

Reviewer #1: Yes

3. Has the statistical analysis been performed appropriately and rigorously? 

Reviewer #1: Yes

4. Have the authors made all data underlying the findings in their manuscript fully available?

Reviewer #1: Yes

5. Is the manuscript presented in an intelligible fashion and written in standard English?

Reviewer #1: Yes

6. Review Comments to the Author

Reviewer #1: (No Response)

7. PLOS authors have the option to publish the peer review history of their article (what does this mean?). If published, this will include your full peer review and any attached files.

Reviewer #1: **Yes: **Erwan David

---

## [Author Response · Author response to Decision Letter 1]

5 May 2022

Dear Editor, Dear Reviewers,

My colleagues and I thank you again very much for your relevant comments and advice on our manuscript entitled “Visual tracking assessment in a soccer-specific virtual environment: a web-based study”. Please find attached the manuscript revised following your comments. The requested responses and additional control analysis have been incorporated into this revised manuscript. The description of data contained in the additional supplementary files has also been added. You will find the changes colored in red in the revised manuscript. All changes are also noted in this rebuttal letter and the lines mentioned refer to the manuscript file with the tracked changes. We also want to inform you that we have removed the web address for the reference 46 (STATS - SportsVU) because it was not functional, and this one was indicated instead:

https://www.statsperform.com/team-performance/football-performance/optical-tracking/

Thank you again for your comments that helped us improve the manuscript.

Kind regards,

Additional Editor Comments:

1. I kindly ask you to also specify and describe, in the “Supporting information” section of the manuscript, what data are contained in supplementary files S3, S4, and S5.

Thank you for the reminder. Here are the added descriptions:

“S1 File. The mean ratio of successful visual tracking by participants in both experimental conditions.”

“S2 File. The raw dataset (before the data is centered-reduced) used for the clustering of target-players in STRU condition. Each row represents an individual target-player and each column represents a situational feature (columns 2-9) or a visual tracking ratio (columns 10-12).”

“S3 File. The raw dataset (before the data is centered-reduced) used for the clustering of target-players in UNSTRU condition.”

The supporting files were rearranged in ascending numerical order according to their first appearance in the manuscript: 

S1 File was first cited on line 293.

S2 File and S3 File were first cited on line 326.

S4 File and S5 File were first cited on line 352.

2. Reviewer 2 Point 3a: In your response to this reviewer comment, you present a table showing tracking performance as function of soccer viewing frequency. You state that even though “mean visual tracking performance appeared to increase with the frequency of soccer game viewing, but we assumed that this effect was not significant due to the large standard deviation”. I have plotted the data you presented, and computed 95% confidence intervals from your reported standard deviations and sample sizes. From these, it looks like there could be a significant effect, as in a few instances the means and confidence intervals of tracking performance across soccer viewing frequency do not overlap. Whichever the case, I suggest it is always best to verify ones’ assumptions when possible. I thus ask you to explicitly test whether a statistical statistically significant effect exists in these data. Further, I believe the reviewer’s main question was whether the observed differences between study groups could be explained by soccer viewing frequency. Thus it would be useful to test and report whether the three study groups differed significantly in soccer viewing frequency (e.g. by running a one-way ANOVA on viewing frequency, with sport practice as the between-subjects main effect. If the groups differ in viewing frequency the same way they differ in tracking performance, then this potential confound should be included and discussed in the discussion section of the manuscript. Moreover, when a reviewer brings up a question, it is good practice to address it both in the response letter as well as in the main manuscript, since future readers may have the same question. I thus ask you to incorporate your response to this reviewer comment in the main manuscript. This could be simply a few lines in the methods or the discussion section of the manuscript, where you point out the issue and report the result of your control analyses (once you have performed them).

Thank you for the insightful comment. 

This control analysis has been added in the discussion of the manuscript (lines 443-463) :

“It seems likely that people who regularly watch soccer games would also exhibit enhanced visual tracking abilities in situations with soccer-specific trajectories, even if they do not necessarily play soccer themselves. In this regard, participants were asked to indicate how often they watched soccer games on a 5-point Likert scale (Never - Once a year - Once a month - Once a week - Several times a week) in the online form prior to performing the visual tracking task. The results of a two-way mixed ANOVA revealed a significant effect of condition (F(1,45)=28.170, p<0.001) but no effect of viewing frequency (F(4,45)=0.653, p=0.628) and no interaction effect between these two factors (F(4,45)=0.647, p=0.631) on visual tracking ratio. It appears that participants with higher viewing frequency of soccer games did not show better visual tracking performance, whether faced with situations with soccer-specific trajectories or not. In addition, the results of a one-way ANOVA revealed a significant main effect of the group on viewing frequency (F(2)=18.501, p<0.001). In pairwise comparisons, a higher viewing frequency was observed in SOCC than in TEAM (p<0.001), a higher viewing frequency was observed in SOCC than in NoTEAM (p<0.001), but no differences were observed between TEAM and NoTEAM (p=1.000). Thus, the groups did not differ equally in viewing frequency of soccer games and visual tracking performance. Overall, the difference in visual tracking performance between participants in situations with soccer-specific trajectories cannot be explained solely by whether or not they regularly watch soccer games. What is important seems to rely more on the viewing perspective than on the viewing frequency.”

3. Reviewer 2 point 3b: As above, I ask you to report the result of these control analyses in the main manuscript.

Thank you for the comment. As requested, the analysis has been added in the manuscript (lines 463-476):

“Attentional demands in the field differ depending on the playing position. Differences in visual tracking performance have been observed between frontcourt and backcourt basketball players [16], and between frontward and backwards rugby players [19]. 

The defender's viewpoint was chosen in this study because most teammates and opponents can be seen within a maximum visual angle of 180°, which increased the crowding of the field of view by the players. With a more advanced viewpoint on the field, the players to be monitored would have been distributed 360° around the observer. A similar defender viewpoint was used in the study by Roca and colleagues (2013) to investigate participants' anticipation when viewing videos of soccer game situations [24]. It can be assumed that defenders would perform better than strikers in the current investigation. It would be interesting to see if playing position would have also influenced tracking performance in this study, but the sample size of soccer players group did not allow for this statistical analysis (3 strikers, 6 midfielders, 6 defenders, and 1 participant did not fill in his/her playing position).”

4. Reviewer 2 point 3d: I suggest you should include in the manuscript these considerations and control analyses regarding frame rate and screen resolution. Perhaps you should also specify that by asking participants to position their eyes at a distance from the screen equivalent to its width, the screen should have subtended approximately 53 degrees of visual angle for all participants. For future reference, you should be aware that there exist validated methods to control viewing distance/stimulus size for online experiments, e.g.: 

Li, Q., Joo, S. J., Yeatman, J. D., & Reinecke, K. (2020). Controlling for participants’ viewing distance in large-scale, psychophysical online experiments using a virtual chinrest. Scientific Reports, 10(1), 1-11. 

Lago, M. A. (2021). SimplePhy: An open-source tool for quick online perception experiments. Behavior Research Methods, 53(4), 1669-1676

Thank you for the comment and the references to methods we were not aware of.

The suggestion has been added in the methods section (lines 221-222):

“In this way, the screen should have subtended approximately 53 degrees of visual angle for all participants. “

These considerations and control analyses have been added in the manuscript (lines 493-512):

“This instruction was made to ensure standardization of viewing conditions, as visual tracking performance can be influenced by the tracking area size and the speed of targets [54]. For future online investigations, validated methods are now available to control viewing distance and stimulus size [55, 56]. Regarding the stimulus size issue in the current investigation, the task could only be performed in full screen mode. The width and height of the browser viewport of participants have been recorded to control the influence of screen dimension on visual tracking performance. The height of the windows varied from 578 to 1137 pixels and the width of windows varied from 960 to 1920 pixels. Pearson’s correlation between mean tracking performance and screen height was 0.141 (p=0.425). Pearson’s correlation between mean tracking performance and screen width was -0.360 (p=0.306). Therefore, participants' visual tracking performance is assumed not to have been influenced by the size of the participants' computer screen. However, the participants' browser frame rate was not recorded. Latency may have influenced the speed of objects and disrupted the fluidity of movement during the task. Although participants did not complain about latency issues, it can only be assumed without guarantee that few or no latencies occurred during the experiment. During pre-tests with different computers, browsers and networks, no latency has been experienced. The data required to execute a trial was loaded into the browser's cache before the trial began, which avoided network latency as the virtual players moved. ”

The suggested references have been added to the manuscript and the list of references. Also, this reference has been added to the list of references:

Franconeri SL, Lin JY, Enns JT, Pylyshyn ZW, Fisher B. Evidence against a speed limit in multiple-object tracking. Psychonomic bulletin & review. 2008;15(4):802–808

5. Reviewer 2 point 3e: Please add a few short sentences in the methods section of the manuscript explaining your reasoning in selecting the number of trials.

Thank you for the comment, the explanation has been added in the methods section of the manuscript (lines 213-217):

“The number of trials (15 per condition) may seem small compared to other investigations on visual tracking expertise [17, 19], but it was intended to encourage participants to remain focused until the end of the experiment. In addition, inter-individual differences in visual tracking performance can appear even with a very low number of trials per condition [18].”

6. Reviewer 2 point 3f: please include these considerations in the discussion section of the manuscript.

Thank you for the comment. In this regard, the following paragraph has been added in the discussion section of the manuscript (lines 477-488):

“The ball and the ball carrier are important visual cues that soccer players must focus on to anticipate the outcome of a game situation [24]. North and colleagues (2017) observed that soccer players were more accurate in recognizing familiar soccer situations presented in point light displays when the ball was present than when it was not [53]. In situations with soccer-specific trajectories, the presence of the ball would probably have helped to perceive structured collective behaviors and thus the visual tracking of several players. However, it was difficult to estimate how the presence of the ball would have influenced participants' visual tracking in situations with pseudo-random trajectories. Thus, the ball was removed from the entire experiment because its presence would surely have affected participants' visual tracking performance differently between the two experimental conditions. The influence of ball presence on the visual tracking in situations with soccer-specific trajectories should be studied in future research.”

This reference has been added to the list of references:

North, J. S., Hope, E., & Williams, A. M. (2017). The role of verbal instruction and visual guidance in training pattern recognition. Frontiers in psychology, 8, 1473.

7. Line 531: “Abrupt changes … were observed”

Thank you for this comment, the error has been corrected.

---

## [Editor Report · Decision Letter 2]

25 May 2022

Visual tracking assessment in a soccer-specific virtual environment: a web-based study

PONE-D-21-39830R2

Dear Dr. Vu,

We’re pleased to inform you that your manuscript has been judged scientifically suitable for publication and will be formally accepted for publication once it meets all outstanding technical requirements.

Kind regards,

Guido Maiello

Academic Editor

PLOS ONE
---

## [Editor Report · Acceptance letter]

1 Jun 2022

PONE-D-21-39830R2 

Visual tracking assessment in a soccer-specific virtual environment: a web-based study 

Dear Dr. Vu:

I'm pleased to inform you that your manuscript has been deemed suitable for publication in PLOS ONE. Congratulations! Your manuscript is now with our production department. 

Kind regards, 

on behalf of

Dr. Guido Maiello 

Academic Editor

PLOS ONE